# Efficient Generative Modeling with Residual Vector Quantization-Based Tokens

Jaehyeon Kim [*1†]   Taehong Moon [*2]   Keon Lee [2]   Jaewoong Cho [2]

## Abstract

We introduce ResGen, an efficient Residual Vector Quantization (RVQ)-based generative model for high-fidelity generation with fast sampling. RVQ improves data fidelity by increasing the number of quantization steps, referred to as depth, but deeper quantization typically increases inference steps in generative models. To address this, ResGen directly predicts the vector embedding of collective tokens rather than individual ones, ensuring that inference steps remain independent of RVQ depth. Additionally, we formulate token masking and multi-token prediction within a probabilistic framework using discrete diffusion and variational inference. We validate the efficacy and generalizability of the proposed method on two challenging tasks across different modalities: conditional *image generation* on ImageNet 256×256 and zero-shot *text-to-speech synthesis*. Experimental results demonstrate that ResGen outperforms autoregressive counterparts in both tasks, delivering superior performance without compromising sampling speed. Furthermore, as we scale the depth of RVQ, our generative models exhibit enhanced generation fidelity or faster sampling speeds compared to similarly sized baseline models.

## 1. Introduction

Recent advancements in deep generative models have shown significant success in high-quality, realistic data generation across multiple domains, including language modeling (Achiam et al., 2023; Touvron et al., 2023; Reid et al., 2024), image generation (Rombach et al., 2022; Saharia et al., 2022; Betker et al., 2023), and audio synthesis (Wang et al., 2023; Shen et al., 2024; Rubenstein et al., 2023). While these models have demonstrated remarkable success, particularly with the effective scaling with both data and model sizes (Kaplan et al., 2020; Peebles & Xie, 2023), challenges remain when aiming for high-fidelity generation, especially in terms of balancing generation quality with computational efficiency. The demand for more detailed, high-resolution outputs such as images (Kang et al., 2023; He et al., 2023), videos (Bar-Tal et al., 2024) and audio (Evans et al., 2024; Copet et al., 2024) has led to the exploration of new approaches that can handle long input sequences and complex data structure effectively (Saharia et al., 2022; Ding et al., 2023).

One promising approach to address these challenges is Residual Vector Quantization (RVQ) (Lee et al., 2022), which improves data reconstruction quality without increasing sequence length. RVQ extends Vector Quantized Variational Autoencoders (VQ-VAEs) (Van Den Oord et al., 2017) by iteratively applying vector quantization to the residuals of previous quantizations. This process results in token sequences that are shorter in length but deeper in hierarchy, effectively compressing data while maintaining high reconstruction fidelity. However, despite the advantages of RVQ in data compression, generative modeling on RVQ-based token sequences introduces new challenges. The hierarchical depth of these token sequences complicates the modeling process, particularly for autoregressive models whose sampling steps typically scale with the product of sequence length and depth (Lee et al., 2022). Although non-autoregressive approaches have been explored along either sequence length or depth (Borsos et al., 2023; Copet et al., 2024; Kim et al., 2024), existing methods do not effectively eliminate the sampling complexity associated with both dimensions simultaneously.

In this paper, we present ResGen, an efficient RVQ-based generative modeling designed to achieve high-fidelity sample quality *without compromising sampling speed*. Our key idea lies in the direct prediction of vector embeddings of collective tokens rather than predicting each token individually. By predicting cumulative embeddings, the model captures correlations among consecutive tokens across depths. This approach allows us to decouple inference steps from both

---

[*]Equal contribution. [†]Work done at KRAFTON. [1]NVIDIA [2]KRAFTON. Correspondence to: Jaewoong Cho <jw-cho@krafton.com>.

*Proceedings of the $42^{nd}$ International Conference on Machine Learning*, Vancouver, Canada. PMLR 267, 2025. Copyright 2025 by the author(s).

sequence length and depth, resulting in a model that generates high-fidelity samples efficiently. Additionally, we extend our approach involving a token masking strategy and a multi-token prediction mechanism within a principled probabilistic framework using a discrete diffusion process and variational inference.

We validate the efficacy and generalizability of ResGen across two real-world generative tasks: conditional image generation on ImageNet 256×256 and zero-shot text-to-speech synthesis. Experimental results demonstrate superior performance over autoregressive counterparts in these tasks. Furthermore, as we scale the depth of RVQ, ResGen exhibits enhanced sampling quality or faster speeds compared to similar-sized baseline generative models. We also analyze model characteristics under varying hyperparameters, such as sampling steps, and examine their impact on generation quality in our ablation study.

The rest of the paper is organized as follows. In Section 2, we provide the background for our study to establish the foundational understanding necessary for the subsequent discussion of our method. In Section 3, we introduce the ResGen framework, detailing the formulation of masked token prediction as a discrete diffusion process and the decoupling of generation iteration from token sequence length and depth. We also compare our approach with previous methods, highlighting the advantages of our strategy. In Section 4, we provide the context for our study with relevant prior work. In Section 5, we present experimental results that validate the performance of ResGen, along with an ablation study on model performance with different RVQ depths and sampling steps. Finally, in Section 6 and Appendix D, we summarize the key findings of the paper, discuss limitations and directions for future research, respectively.

## 2. Background

**Masked Token Modeling.** Masked token modeling, introduced in prior work (Chang et al., 2022), is a generative framework that operates on token sequences derived from the quantized encoder outputs of a Vector Quantized Variational AutoEncoder (VQ-VAE) (Van Den Oord et al., 2017). The core idea involves randomly masking a subset of input tokens and training the model to predict these masked tokens using a cross-entropy loss.

Let $\boldsymbol{x} = (x_1, \ldots, x_L)$ be a token sequence in $\mathbb{N}^L$ and $\boldsymbol{m} = (m_1, \ldots, m_L)$ be a corresponding binary mask, where $m_i \in \{0, 1\}$. By definition $m_i = 0$ indicates that token $x_i$ is masked. We construct a masked token sequence $\boldsymbol{x} \odot \boldsymbol{m}$ by element-wise multiplying $\boldsymbol{x}$ and $\boldsymbol{m}$. The training objective is then formulated as:

$$\mathcal{L}_{\text{mask}}(\boldsymbol{x}, \boldsymbol{m}; \theta) = - \sum_{i \in \{i | m_i = 0\}} \log p_\theta(x_i | \boldsymbol{x} \odot \boldsymbol{m}),$$

where $\theta$ denotes the model parameters and the summation over $i$ includes only those positions where the token is masked, denoted by $m_i = 0$. The masking process involves selecting a number of tokens $n$ to mask, determined by a masking schedule $n = \lceil \gamma(r) \cdot L \rceil$. Here, $r$ indicates the current time step in the unmasking process, normalized to range from zero to one, and $\gamma(\cdot)$ is a pre-defined masking scheduling function that monotonically decreases from one to zero as $r$ increases. During training, $r$ is sampled from a uniform distribution.

In the decoding phase, the model employs an iterative prediction process to progressively fill in the masked sequence. At each iteration, the masking ratio $r$ is updated to linearly increase from zero to one. Starting with an entirely masked token sequence, the model predicts the masked tokens, and a subset of these predicted tokens is selected to be unmasked based on confidence scores calculated through prediction probabilities. The number of tokens to unmask at each iteration is determined by the masking schedule.

**Residual Vector Quantization.** Residual Vector Quantization (RVQ) (Lee et al., 2022) has been employed as a more precise alternative to VQ for quantizing latent vectors in VQ-VAE. While previous VQ-VAEs quantize an input by replacing each encoded vector with the nearest embedding from a codebook, RVQ iteratively applies vector quantization to the residuals of previous quantizations.

Formally, let the output of the encoder in a VQ-VAE at the position $i$ be $\boldsymbol{h}_{i,0}$. The residual vector quantizer maps it to a sequence of quantized tokens $\boldsymbol{x} \in \mathbb{N}^{L \times D}$, where $D$ is the total depth of the RVQ process:

$$x_{i,j} = \underset{v \in \{1, \ldots, V\}}{\arg\min} \|\boldsymbol{h}_{i,j-1} - \boldsymbol{e}(v; j)\|^2$$

$$\boldsymbol{h}_{i,j} = \boldsymbol{h}_{i,j-1} - \boldsymbol{e}(x_{i,j}; j) \quad \text{for all } j \in [1, D],$$

where $\boldsymbol{e}(v; j)$ is the $v$-th vector embedding from the codebook at depth $j$, and $V$ is the number of embeddings per depth. Here, $x_{i,j}$ represents the selected embedding index for the $i$-th token at depth $j$, and $\boldsymbol{h}_{i,j}$ denotes the residual vector after the $j$-th quantization step.

The final reconstructed vector is obtained by summing the embeddings across all depths, $\boldsymbol{z}_i = \sum_{j=1}^{D} \boldsymbol{e}(x_{i,j}; j)$. This iterative quantization process enables RVQ to produce a quantized output that closely approximates the original encoder output by increasing the depth $D$ of quantization steps. As a result, RVQ effectively captures the most significant features in the lower quantization layers, while finer details are progressively captured in higher layers.

**Forward masking process**

*Figure 1.* An overview of the forward masking and reverse unmasking processes is shown at the top, with a detailed depiction of the reverse unmasking process below. In the top figure, forward masking proceeds from right to left, incrementally masking more tokens, while reverse unmasking progresses from left to right, iteratively revealing the masked tokens. White boxes denote masked tokens and colored boxes represent tokens that have been uncovered. The bottom figure illustrates the reverse unmasking process in detail. Starting from masked residual vector quantization (RVQ) tokens, our method first predicts cumulative RVQ embeddings. These embeddings are then quantized and partially masked again. Through a series of iterations, each round predicts the values of the masked tokens and replaces them until the entire token sequence is filled.

## 3. Method

In this section, we introduce our method, ResGen, which iteratively fills tokens in a coarse-to-fine manner to achieve efficient and high-fidelity generative modeling with Residual Vector Quantization (RVQ). We structure our discussion into three main parts:

- We present a token masking strategy tailored for RVQ tokens and describe how we model masked token prediction by predicting sum of residual vector embeddings to decouple the generation iterations from the length and depth of token sequences.

- We show that our proposed token masking and multi-token prediction method can be formulated within a probabilistic framework using a discrete diffusion process and variational inference.

- We detail the training and sampling techniques of ResGen, focusing on the implementation of the mixture of Gaussians for latent embedding estimation and enhanced sampling strategies based on model confidence scores.

### 3.1. Masking and Prediction Task Design for RVQ Tokens

**Token Masking for RVQ Tokens.** Our masking strategy progressively masks tokens starting from the highest quantization layers, capitalizing on the hierarchical nature of RVQ where tokens at greater depths capture finer details.

Given a token sequence from RVQ, $\boldsymbol{x} \in \mathbb{N}^{L \times D}$, with sequence length $L$ and depth $D$, we apply a binary mask $\boldsymbol{m} \in \{0,1\}^{L \times D}$, where each $m_{i,j}$ indicates whether the token $x_{i,j}$ is masked ($m_{i,j} = 0$) or not ($m_{i,j} = 1$). The total number of tokens to mask is determined by a masking schedule, $n = \lceil \gamma(r) \cdot L \cdot D \rceil$. Here, $r$ indicates the current time step in the unmasking process, normalized to range from zero to one, and $\gamma(\cdot)$ is a pre-defined masking scheduling function that monotonically decreases from one to zero as $r$ increases. During training, $r$ is sampled from a uniform distribution.

To distribute the $n$ masked tokens across the $L$ positions, the number of tokens to mask at each position $i$, denoted by $k_i$, is sampled without replacement from a multinomial distribution with equal probability across all positions, ensuring that $\sum_{i=1}^{L} k_i = n$. At each position $i$, $k_i$ tokens are masked starting from the highest depth $j = D$ and moving towards lower depths. This ensures that finer details captured at higher depths are masked before coarser information at lower depths, as illustrated in Figure 1.

**Multi-Token Prediction of Masked Tokens.** We describe the training and decoding phases of our multi-token prediction strategy, which efficiently predicts masked tokens by focusing on predicting the aggregated vector embeddings $\boldsymbol{z}$ of collective tokens rather than the individual tokens $\boldsymbol{x}$.

**Training:** Given the input sequence $\boldsymbol{x}$ and the corresponding mask $\boldsymbol{m}$, the model predicts the sum of masked embeddings $\boldsymbol{z}$ such that $\boldsymbol{z}_i = \sum_j \boldsymbol{e}(x_{i,j}; j) \odot (1 - m_{i,j})$ rather

than the target tokens directly, where $e(v; j)$ denotes the $v$-th vector embedding from the RVQ codebook at depth $j$. The training objective is to maximize the log-likelihood of the sum of masked embeddings:

$$\mathcal{L}_{\text{mask}}(\boldsymbol{x}, \boldsymbol{m}; \theta) = - \sum_{i \in \{i | \sum_j m_{i,j} < D\}} \log p_\theta(\boldsymbol{z}_i | \boldsymbol{x} \odot \boldsymbol{m}),$$
(1)

where $\theta$ represents the model parameters and the summation over $i$ includes only those positions where at least one token is masked, denoted by $\sum_j m_{i,j} < D$. To model the distribution $p_\theta$, we employ a mixture of Gaussian distributions. We modify the training objective to encourage the mixture component usage of the mixture of Gaussian distributions, which is described in Appendix A.2.

This method avoids imposing conditional independence of tokens along the depth, which could harm model performance. Instead, it relies on the key idea that accurately predicting the vector embedding $\boldsymbol{z}_i$ is more critical than predicting the individual tokens $\boldsymbol{x}_i$, as the decoder of a VQ-VAE operates on vector embeddings.

**Sampling:** In the decoding phase, the model employs an iterative prediction process to progressively fill in the masked sequence. At each iteration, the masking ratio $r$ is updated to linearly increase from zero to one. Starting with an entirely masked token sequence, the model progressively fills in the sequence in a coarse-to-fine manner. At each step, the model predicts the cumulative masked token embedding $\boldsymbol{z}_i$. These predicted vectors are then quantized into tokens via RVQ quantization. A subset of these predicted tokens is randomly selected to be unmasked, where the number of tokens to unmask at each step is determined by the masking schedule. Although the quantization step at each sampling iteration involves sequential operations to reconstruct tokens from embeddings, it adds negligible overhead compared to the model forward pass.

We summarize the training and sampling algorithms for ResGen in Algorithm 1 and 2 of Appendix.

### 3.2. Formulation within a Probabilistic Framework

We now cast our masked token prediction procedure into a probabilistic framework based on a discrete diffusion model and variational inference. This perspective allows us to view our method as a likelihood-based generative process and provides a theoretical foundation for its design.

**Forward Discrete Diffusion Process.** Consider the token masking process described in Section 3.1. We can interpret this process as the forward diffusion step of a discrete diffusion model on token sequences. The idea is to gradually transform a fully unmasked token sequence $\boldsymbol{x}^{(0)}$ into a fully masked sequence $\boldsymbol{x}^{(T)}$ by progressively increasing the

number of masked tokens at each step.

The masking at each step $t$ is governed by a discrete random process that determines how many tokens to mask. Let $k_i^{(t+1)}$ denote the number of tokens to be newly masked at the next step $t + 1$ for the $i$-th position in a token sequence. The total number of tokens to be masked at step $t + 1$ is $n^{(t+1)} = \sum_{i=1}^L k_i^{(t+1)}$, where $L$ and $D$ are the length and the depth of the token sequence. The probabilistic mechanism is that at each step $t$, we sample the vector $\boldsymbol{k}^{(t+1)} = (k_1^{(t+1)}, ..., k_L^{(t+1)})$ from a multivariate hypergeometric distribution, which corresponds to drawing $n^{(t+1)}$ tokens without replacement from the pool of currently unmasked tokens. Formally, if at step $t$ there remain $LD - \sum_{\tau=1}^t n^{(\tau)}$ unmasked tokens in $\boldsymbol{x}^{(t)}$, then drawing $n^{(t+1)}$ tokens to mask can be modeled as:

$$q(\boldsymbol{k}^{(t+1)} | \boldsymbol{x}^{(t)}) = \frac{\prod_{i=1}^L \binom{D - \sum_{\tau=1}^t k_i^{(\tau)}}{k_i^{(t+1)}}}{\binom{LD - \sum_{\tau=1}^t n^{(\tau)}}{n^{(t+1)}}}.$$

Once we have sampled $\boldsymbol{k}^{(t+1)}$, we construct $\boldsymbol{x}^{(t+1)}$ from $\boldsymbol{x}^{(t)}$ by masking out the newly selected tokens. Specifically, let $\phi$ denote the masked token. Then, resulting masked tokens at each sequence position $i$ are defined as:

$$x_{i,j}^{(t+1)} = \begin{cases} x_{i,j}^{(t)} & \text{if } j \leq D - \sum_{\tau=1}^t k_i^{(\tau)} \\ \phi & \text{otherwise} \end{cases}.$$

An attractive property of this forward diffusion process is that we can write closed-form expressions for both the marginal distributions and conditional distributions at intermediate steps. Since the forward process is defined by incremental masking without replacement, we can directly integrate over all intermediate steps. This yields the marginal distribution of $\boldsymbol{x}^{(t)}$ given $\boldsymbol{x}^{(0)}$:

$$q(\boldsymbol{x}^{(t)} | \boldsymbol{x}^{(0)}) = \frac{\prod_{i=1}^L \binom{D}{\sum_{\tau=1}^t k_i^{(\tau)}}}{\binom{LD}{\sum_{\tau=1}^t n^{(\tau)}}},$$

which expresses the probability of having $\sum_{\tau=1}^t k_i^{(\tau)}$ tokens masked in each segment $i$, given that a total of $\sum_{\tau=1}^t n^{(\tau)}$ tokens have been masked overall up to step $t$.

Similarly, we can write the conditional distribution of $\boldsymbol{x}^{(t)}$ given $\boldsymbol{x}^{(t+1)}$ and $\boldsymbol{x}^{(0)}$:

$$q(\boldsymbol{x}^{(t)} | \boldsymbol{x}^{(t+1)}, \boldsymbol{x}^{(0)}) = \frac{\prod_{i=1}^L \binom{\sum_{\tau=1}^{t+1} k_i^{(\tau)}}{k_i^{(t+1)}}}{\binom{\sum_{\tau=1}^{t+1} n^{(\tau)}}{n^{(t+1)}}},$$

reflecting the probability of having arrived at $\boldsymbol{x}^{(t)}$ from $\boldsymbol{x}^{(t+1)}$ by considering how many tokens were masked in the last step. In this sense, the forward and backward processes are fully characterized by the combinatorial structure of drawing tokens without replacement.

**Reverse Discrete Diffusion Process.** The reverse process aims to reconstruct the original tokens from partially masked sequences. Given the model's probability to reconstruct the original tokens $p_\theta(\boldsymbol{x}^{(0)} \mid \boldsymbol{x}^{(t+1)})$, the probability to reverse a diffusion step $p_\theta(\boldsymbol{x}^{(t)} \mid \boldsymbol{x}^{(t+1)})$ is defined as:

$$\sum_{\boldsymbol{x}^{(0)}} q(\boldsymbol{x}^{(t)} \mid \boldsymbol{x}^{(t+1)}, \boldsymbol{x}^{(0)}) p_\theta(\boldsymbol{x}^{(0)} \mid \boldsymbol{x}^{(t+1)}).$$

This formulation lets us compute the variational lower bound of the data log-likelihood:

$$\log p_\theta(\boldsymbol{x}^{(0)}) \geq -\mathbb{E}_q\left[\mathcal{L}_T + \sum_{t \geq 1} \mathcal{L}_t + \mathcal{L}_0\right],$$

$$\text{where} \quad \mathcal{L}_T = D_{\mathrm{KL}}\Big(q(\boldsymbol{x}^{(T)} \mid \boldsymbol{x}^{(0)}) \,\|\, p(\boldsymbol{x}^{(T)})\Big),$$

$$\mathcal{L}_t = D_{\mathrm{KL}}\Big(q(\boldsymbol{x}^{(t)} \mid \boldsymbol{x}^{(t+1)}, \boldsymbol{x}^{(0)}) \,\|\, p_\theta(\boldsymbol{x}^{(t)} \mid \boldsymbol{x}^{(t+1)})\Big),$$

$$\text{and} \quad \mathcal{L}_0 = -\log p_\theta(\boldsymbol{x}^{(0)} \mid \boldsymbol{x}^{(1)}).$$

Here, $\mathcal{L}_T$ is the prior loss, which becomes zero since $\boldsymbol{x}^{(T)}$ is fully masked, $\mathcal{L}_t$ are the diffusion losses at each step $t$, and $\mathcal{L}_0$ is the reconstruction loss. By combining the diffusion and the reconstruction losses while placing equal emphasis on predicting the original tokens at each step, we can derive a simplified loss function:

$$\mathcal{L}_{\mathrm{simple}}(\boldsymbol{x}^{(0)}; \theta) = -\log p_\theta(\boldsymbol{x}^{(0)} \mid \boldsymbol{x}^{(t)}). \tag{2}$$

**Latent Modeling with Variational Inference.** To efficiently handle dependencies across token depths, we adopt a multi-token prediction approach inspired by CLaM-TTS (Kim et al., 2024). Instead of predicting tokens individually, we predict their cumulative vector embedding $\boldsymbol{z}$. This approach aligns naturally with the RVQ dequantization process and decouples the generation time complexity from the token depth.

We use variational inference to derive the upper bound of the negative log-likelihood $-\log p_\theta(\boldsymbol{x}^{(0)} \mid \boldsymbol{x}^{(t)})$:

$$\mathbb{E}_{q_{\boldsymbol{z}}}\left[-\log p(\boldsymbol{x}^{(0)}|\boldsymbol{z}, \boldsymbol{x}^{(t)}) - \log \frac{p_\theta(\boldsymbol{z} \mid \boldsymbol{x}^{(t)})}{q(\boldsymbol{z} \mid \boldsymbol{x}^{(0)}, \boldsymbol{x}^{(t)})}\right].$$

Assuming $p(\boldsymbol{x}^{(0)}|\boldsymbol{z}, \boldsymbol{x}^{(t)})$ corresponds to RVQ quantization and $q(\boldsymbol{z} \mid \boldsymbol{x}^{(0)}, \boldsymbol{x}^{(t)})$ to RVQ dequantization of the masked tokens, we focus on the remaining term:

$$\mathcal{L}_{\mathrm{mask}}(\boldsymbol{x}^{(0)}, \boldsymbol{x}^{(t)}; \theta) = -\log p_\theta(\boldsymbol{z}|\boldsymbol{x}^{(t)}),$$

which matches the prediction loss in Equation 1.

## 4. Related Work

In this work, we refer to discrete diffusion models (DDMs) as a class of generative models that learn to reverse a defined corruption process applied to discrete data, such as sequences of tokens (Austin et al., 2021). This typically involves iteratively refining corrupted tokens or progressively unmasking masked tokens. A prominent strategy within this framework is masked generative modeling, also known as masked diffusion or masked token modeling, where the corruption process specifically involves masking portions of the token sequence, and the model learns to predict the content of these masked positions (Gu et al., 2022; Chang et al., 2022). Models such as VQ-Diffusion (Gu et al., 2022) and, conceptually, MaskGIT (Chang et al., 2022) exemplify this masked diffusion approach for generating flat token sequences, leading to improved sampling efficiency over autoregressive models. GIVT (Tschannen et al., 2024) introduces a method that replaces softmax-based token prediction with mixture-of-Gaussians-based vector prediction in masked token prediction, progressively filling masked positions with predicted vectors. Build upon principles similar to MaskGIT, MAGVIT-v2 (Yu et al., 2024) and MaskBit (Weber et al.) have demonstrated strong performance by incorporating innovations such as improved quantization schemes, notably Lookup-Free Quantization (LFQ) (Yu et al., 2024), which represents each token as a collection of bits. MaskBit further advances this by grouping the bits of each token and leveraging the partially unmasked bits to predict the masked portions.

Separate from the discrete diffusion approaches defined above, recent works like VAR (Tian et al., 2024), MAR (Li et al., 2024) and HART (Tang et al., 2024) propose alternative paradigms to token-based autoregressive modeling. VAR introduces a coarse-to-fine next-scale prediction mechanism, effectively capturing hierarchical structures in images; for a detailed discussion of the distinctions between VAR's hierarchical modeling and ResGen's approach, particularly concerning representation, generation, and adaptability, please see Appendix C.1. MAR eliminates the reliance on discrete tokens by modeling probabilities in continuous-valued space using a diffusion-based approach, simplifying the pipeline while maintaining strong performance. HART proposes a hybrid approach that decomposes the latent space into discrete tokens, modeled autoregressively, and continuous residuals, modeled with a diffusion process. This strategy is designed to reduce the number of sampling steps compared to fully continuous methods like MAR.

However, these methods primarily deal with flat token sequences and do not consider the hierarchical depth inherent in RVQ. RQ-Transformer (Lee et al., 2022) was the first to demonstrate generative modeling on RVQ tokens using an autoregressive model over the product of sequence length and depth, resulting in increased computational complexity. Vall-E (Wang et al., 2023) predicts the tokens at the first depth autoregressively and then predicts the remaining tokens at each depth in a single forward pass sequentially. SoundStorm (Borsos et al., 2023) generates tokens using

masked token prediction given semantic tokens, but still has sampling time complexity that increases linearly with the residual quantization depth. NaturalSpeech 2 (Shen et al., 2024) employs diffusion-based generative modeling in the RVQ embedding space instead of token generation. CLaM-TTS (Kim et al., 2024) employs vector prediction for multi-token prediction but operates in an autoregressive manner along the sequence length.

In contrast to these approaches, our method offers a more efficient solution for generative modeling with RVQ tokens. We propose a strategy that predicts the vector embedding of masked tokens, decoupling the sampling time complexity from both sequence length and token depth. By predicting cumulative vector embeddings rather than individual tokens, our method efficiently handles the hierarchical structure of tokens, offering enhanced sampling efficiency and high-fidelity generation.

# 5. Experiments

In this section, we demonstrate the effectiveness of our approach in both image generation and text-to-speech synthesis, highlighting its generalizability and efficiency.

## 5.1. Experimental Setting

**Experiment Tasks.** For the vision domain, we focus on conditional image generation tasks on ImageNet (Krizhevsky et al., 2017) at a resolution of $256 \times 256$. In the audio domain, we evaluate our model using two tasks inspired by Voicebox (Le et al., 2023): 1) *continuation*: given a text and a 3-second segment of ground truth speech, the goal is to generate seamless speech that continues in the same style as the provided segment; 2) *cross-sentence*: given a text, a 3-second speech segment, and its transcript (which differs from the text), the objective is to generate speech that reads the text in the style of the provided segment.

**Evaluation Metrics.** For vision tasks, we employ the Fréchet Inception Distance (FID) (Heusel et al., 2017) for comparing it with other state-of-the-art image generative models. We set the sample size for FID calculation to 50K in all experiments. For audio tasks, we evaluate the models using the following objective metrics: Character Error Rate (CER), Word Error Rate (WER), and Speaker Similarity (SIM), as described in VALL-E (Wang et al., 2023) and CLaM-TTS (Kim et al., 2024). CER and WER measure the intelligibility and robustness. For SIM, we adopt SIM-o and SIM-r metrics from Voicebox (Le et al., 2023). SIM-o evaluates the similarity between the generated speech and the original target speech, while SIM-r assesses the similarity between the target speech and its reconstruction, which is obtained by processing the original speech through a pre-trained autoencoder and vocoder.

**Baselines and Training Configurations.** In the vision domain, we compare our models with recent generative model families, including (1) *autoregressive models*: RQ-transformer (Lee et al., 2022), VAR (Tian et al., 2024), MAR (Li et al., 2024); and (2) *non-autoregressive models*: MaskGIT (Chang et al., 2022), DiT (Peebles & Xie, 2023). For the audio task, we benchmark the proposed model against state-of-the-art TTS models, including (1) *autoregressive models*: VALL-E (Wang et al., 2023), SPEAR-TTS (Kharitonov et al., 2023), and CLaM-TTS (Kim et al., 2024); and (2) *non-autoregressive models*: YourTTS (Casanova et al., 2022), VoiceBox (Le et al., 2023), and DiTTo-TTS (Lee et al., 2025). The training configurations for our models are detailed in Appendix A.1.

## 5.2. Experimental Results

### 5.2.1. VISION TASK: ABLATION STUDIES

To validate the effectiveness of our method as a generative model for RVQ tokens, we compare it with two baseline models, MaskGIT and RQ-Transformer, using an identical RVQ representation with depth 16. MaskGIT, originally designed to predict randomly masked tokens at a single depth, is adapted to a multi-depth setting by autoregressively generating tokens depth by depth, conditioning each prediction on the previously generated depths.

We therefore investigate three generation strategies for $8 \times 8 \times 16$ RVQ tokens: (i) *fully autoregressive* (RQ-Transformer); (ii) *masked sequence + autoregressive depth* (our MaskGIT variant); and (iii) *fully masked* (ResGen). To ensure a fair comparison, we configured each model with comparable parameter counts: ResGen (576 M), RQ-Transformer (626 M), and MaskGIT (580 M), and trained them for 2.8 M steps.

As summarized in the left Table 1, ResGen shows higher generation quality than both baselines while requiring fewer inference steps than RQ-Transformer. The relatively lower performance of the depth-wise MaskGIT variant highlights the difficulty of extending single-depth masked models to multi-depth RVQ tokens, reinforcing our design choice of predicting all depths jointly.

Beyond these baseline comparisons, further experiments, also detailed in the left Table 1, explore key design choices and the scalability of ResGen. First, we investigate an architectural variant of ResGen that predicts discrete tokens directly and in parallel across all depths at each step, using the loss function defined in Equation 2. This variant yields competitive results relative to the baselines but is outperformed by the final version of ResGen, highlighting the effectiveness of the cumulative embedding prediction strategy. Second, to assess model scalability, we increase the parameters of ResGen from its 576 M version to a 1 B pa-

*Table 1.* Ablation study on ResGen (Ours). The left table compares image generation quality and efficiency among ResGen, RQ-transformer, and MaskGIT, evaluated using the same RVQ tokens. The right table reports efficiency of AR-ResGen using 2, 4, and 8 iterative refinement steps for depth prediction, compared with RQ-Transformer. The boldface indicates the best result. Wall-clock time results reflect the time required to generate a single sample on an NVIDIA A100 GPU.

| Model | FID (w/o CFG) ↓ | FID (w/ CFG) ↓ | Steps |
|---|---|---|---|
| RQ-transformer | 15.71 | 5.50 | 1024 |
| MaskGIT | 28.40 | 9.83 | 63 |
| ResGen | **8.77** | **2.43** | 63 |
| *Architectural and Scaling Analysis* | | | |
| ResGen (Direct Token Prediction) | 12.79 | 2.91 | 63 |
| ResGen (1B Parameters) | **8.12** | **2.26** | 63 |

| Model | FID (w/o CFG) ↓ | FID (w/ CFG) ↓ | Wallclock Time |
|---|---|---|---|
| RQ-transformer | **15.71** | 5.50 | 5.35s |
| AR-ResGen (1 step) | 27.45 | 6.47 | 1.30s |
| AR-ResGen (2 step) | 24.10 | 5.33 | 1.56s |
| AR-ResGen (4 step) | 23.75 | 5.30 | 2.00s |
| AR-ResGen (8 step) | 23.48 | **5.22** | 3.05s |

rameter model. This larger model demonstrates further FID improvements, confirming that ResGen scales effectively with increased capacity.

The right of Table 1 summarizes a stricter ablation designed to disentangle the efficiency gains introduced by our depth modeling strategy from those brought by the masked-generation paradigm. This ablation features AR-ResGen, a variant where the spatial sequence is still generated autoregressively, identical to RQ-Transformer. Crucially, AR-ResGen keeps the overall parameter budget comparable (625 M) and differs from the RQ-Transformer only by replacing its depth transformer with MLP to predict our cumulative embedding.

The data in the table shows that AR-ResGen attains an FID of 5.22 with 8 refinement iterations while reducing the decoding time from 5.35 to 3.05 s (×1.8 faster). Even with just two iterations it reaches an FID of 5.33s in 1.56s, yielding a ×3.4 speed-up over RQ-Transformer at comparable quality. These results isolate and confirm that our depth prediction strategy alone, independent of the sequence generation framework, delivers substantial improvements in both the generation speed and the sample quality. These results demonstrate that ResGen sets a new benchmark for both efficiency and quality in generative modeling.

Finally, Appendix B.2 examines the effect of sampling hyperparameters, specifically the number of steps and temperature scaling, on generation quality, and Appendix C.3 investigates the sensitivity of ResGen's performance to alternative masking strategies.

### 5.2.2. VISION TASK: BENCHMARK COMPARISONS

Next, we assess the effectiveness of our generative modeling by comparing it to existing vision generative model families. We train two variants of ResGen, termed ResGen-rvq8 and ResGen-rvq16, using 8-depth and 16-depth RVQ tokens, respectively, for up to 7 million steps. The results, shown in Table 2, compare these models across three key aspects: generation quality, memory efficiency, and generation speed. Generation quality is assessed using the FID metric, mem-

ory efficiency is evaluated based on the maximum batch size, which refers to the maximum number of latent representations that a generative model can process during inference on the same device, and generation speed is measured as the wall-clock time required to generate a single sample. Detailed sampling hyperparameters can be found in Table 4.

**Generation quality.** Among models with similar parameter counts, and specifically excluding larger models such as VAR-d30 and MAR-H, ResGen-rvq16 demonstrates highly competitive performance, which achieves an FID of 1.93 with classifier-free guidance (CFG). Despite its slightly higher FID score compared to MAR-L, ResGen achieves comparable quality with much faster sampling speed, demonstrating its efficiency in balancing quality and resource usage. These results highlight the competitive generation capabilities of ResGen, which closely rivals the MAR series.

**Speed efficiency.** As shown in Figure 2, ResGen-rvq16 ranks second only to VAR, making it a practical choice for scenarios where speed is critical. Unlike certain models that compromise speed for quality, our method maintains a favorable balance, offering both rapid generation and competitive quality. Compared to the MAR series, which achieves marginally better FID scores, ResGen's superior speed positions it as an efficient solution for real-time or high-throughput applications.

**Memory efficiency.** ResGen consistently exhibits superior efficiency, achieving the largest latent batch size. Specifically, ResGen-rvq16 supports a maximum latent batch size of 1,915, surpassing MAR-B's capacity of 1,738. This efficiency enables large-scale generative tasks and mitigates computational bottlenecks in production pipelines.

**Comparison with recent masked generative models.** To provide a broader context for ResGen's performance, we also consider recent state-of-the-art masked generative models like MAGVIT-v2 (Yu et al., 2024) and MaskBit (Weber et al.). As detailed in Table 8, these models achieve strong FID scores (MAGVIT-v2: 1.78, MaskBit: 1.62), which

*Table 2.* Comparison of generative models on class-conditional ImageNet at a resolution of 256×256. Boldface denotes the best result and underline indicates the second best; an asterisk (*) marks scores reported in their original papers. For RQ-Transformers, FID scores presented in the 'FID (w/ CFG)' column are based on results obtained with rejection sampling. 'Code length' refers to the sequence length of the latent representations.

| Model | Code length | Params | FID (w/o CFG) ↓ | FID (w/ CFG) ↓ | Maximum batch size ↑ |
|---|---|---|---|---|---|
| MaskGIT | 256 | 277M | 6.18* | - | - |
| DiT-XL/2 | 256 | 675M | 9.62* | 2.27* | 1159 |
| VAR-d16 | 256 | 310M | 12.18 | 3.30* | 247 |
| VAR-d20 | 256 | 600M | 8.60 | 2.57* | 148 |
| VAR-d24 | 256 | 1.0B | 6.43 | 2.09* | 102 |
| VAR-d30 | 256 | 2.0B | 5.31 | 1.92* | 60 |
| MAR-B | 256 | 208M | 3.48* | 2.31* | 1738 |
| MAR-L | 256 | 479M | 2.60* | 1.78* | 1167 |
| MAR-H | 256 | 943M | **2.35*** | **1.55*** | 812 |
| RQ-Transformer | 64 | 1.4B | 8.71* | 3.89* | 1151 |
| RQ-Transformer | 64 | 3.8B | 7.55* | 3.80* | 390 |
| ResGen-rvq8 | 64 | 576M | 6.56 | 2.71 | **1995** |
| ResGen-rvq16 | 64 | 576M | 6.04 | 1.93 | 1915 |

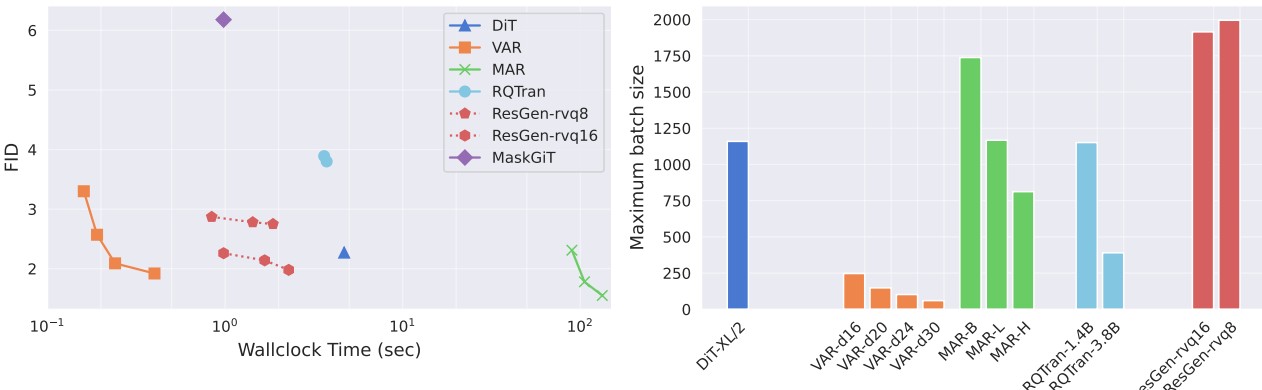

*Figure 2.* The left figure shows the trade-off between sampling speed and generation quality across various generative models. For ResGen, dotted lines indicate performance across different sampling steps, highlighting step-dependent performance improvements. For other models, solid lines connect results corresponding to variations in parameter size. Note that in *ResGen-rvq8* and *ResGen-rvq16*, the number specifies the depth of RVQ. Both wall-clock time and maximum batch size are measured on an NVIDIA A100 GPU.

are currently lower than that of our ResGen-rvq16 (1.93 with CFG). However, ResGen offers distinct architectural advantages, including more effective handling of hierarchical token dependencies and flexible resolution-depth scaling, which enable shorter code lengths. These properties contribute to its strong overall performance and adaptability across diverse settings, as discussed in detail in Appendix C.4.

Interestingly, ResGen-rvq16, with deeper RVQ quantization, achieves better sample quality than ResGen-rvq8 at the same number of sampling steps, with a minimal increase in inference time. This aligns with Appendix A.4, which shows

that greater RVQ depth enhances both reconstruction and generation quality. These results demonstrate the scalability of our approach, showing that deeper RVQ quantization improves performance while maintaining efficiency.

For qualitative evaluation, Figure 5 presents diverse samples generated by the baseline models (VAR, MAR, and DiT) alongside those from ResGen. Additional randomly generated samples from our model are shown in Figure 6. Finally, in Appendix C.2, we present a detailed comparison between ResGen and the faster MAR variant.

*Table 3.* Performances for the *continuation* task (top table) and the *cross-sentence* task (bottom table). The boldface indicates the best result, the underline denotes the second best, and the asterisk denotes the score reported in the baseline paper.

| Model | Params | WER ↓ | CER ↓ | SIM-o ↑ | SIM-r ↑ | Inference Steps ↓ |
|---|---|---|---|---|---|---|
| Ground Truth | n/a | 2.2* | 0.61* | 0.754* | 0.754* | n/a |
| YourTTS | - | 7.57 | 3.06 | 0.3928 | - | **1** |
| Vall-E | 302M | 3.8* | - | 0.452* | 0.508* | - |
| Voicebox | 364M | 2.0* | - | **0.593*** | **0.616*** | 64 |
| CLaM-TTS | 584M | 2.36* | 0.79* | 0.4767* | 0.5128* | - |
| DiTTo-en-L | 508M | 1.85 | 0.50 | 0.5596 | 0.5913 | 25 |
| DiTTo-en-XL | 740M | **1.78*** | **0.48*** | 0.5773* | 0.6075* | 25 |
| Melvae-ResGen | 457M | 1.94 | 0.53 | 0.5421 | 0.5701 | 25 |
| Rvqvae-ResGen | 625M | 1.86 | 0.50 | 0.5853 | 0.5886 | 25 |

| Model | WER ↓ | CER ↓ | SIM-o ↑ | SIM-r ↑ |
|---|---|---|---|---|
| YourTTS | 7.92 (7.7*) | 3.18 | 0.3755 (0.337*) | - |
| Vall-E | 5.9* | - | - | 0.580* |
| SPEAR-TTS | - | 1.92* | - | 0.560* |
| Voicebox | 1.9* | - | **0.662*** | **0.681*** |
| CLaM-TTS | 5.11* | 2.87* | 0.4951* | 0.5382* |
| DiTTo-en-L | 2.69 | 0.91 | 0.6050 | 0.6355 |
| DiTTo-en-XL | 2.56* | 0.89* | 0.627* | 0.6554* |
| Melvae-ResGen | 1.75 | 0.48 | 0.5597 | 0.6061 |
| Rvqvae-ResGen | **1.70** | **0.46** | 0.6037 | 0.6307 |

### 5.2.3. AUDIO TASK

In our Text-to-Speech (TTS) experiments, we compare our method to autoregressive models that generate RVQ tokens, using the same MelVAE module from CLaM-TTS (Kim et al., 2024). As shown in Table 3, our model achieves lower word and character error rates (WER and CER) as well as higher speaker similarity scores (SIM-o and SIM-r) than the baseline and requires fewer inference steps, demonstrating its efficiency in token generation. Notably, our method uses only 25 iterations, which is fewer than the RVQ depth of 32.

We further evaluate our model with deeper RVQ quantization, up to 72 levels, referred to as Rvqvae-ResGen, and compare its performance against recent TTS models. While our results do not surpass state-of-the-art methods such as Voicebox and DiTTo-TTS on every metric, the proposed approach demonstrates substantial advantages in computational efficiency and accuracy. Specifically, ResGen achieves the lowest WER and CER in the *cross-sentence* task, outperforming all baselines. In the *continuation* task, ResGen attains competitive WER and CER scores, ranking second only to DiTTo-en-XL. Furthermore, the use of a deeper RVQ depth enhances leads to improved reconstruction quality and enhanced generation performance. Despite the increased depth, ResGen effectively models these tokens with only 25 inference steps, demonstrating its ability to maintain computational efficiency while delivering high-quality outputs. For qualitative comparison, we present our generated audio samples in the project page[1].

## 6. Conclusion

In this work, we propose ResGen, an efficient RVQ-based discrete diffusion model that generates high-fidelity samples while maintaining fast sampling speeds. By directly predicting the vector embedding of collective tokens, our method mitigates the trade-offs between RVQ depth and inference speed in RVQ-based generative models. We further demonstrate the effectiveness of token masking and multi-token prediction within a probabilistic framework, employing a discrete diffusion process and variational inference. Experimental results on conditional image generation and zero-shot text-to-speech synthesis validate the strong performance of ResGen, which performs comparably to or exceeds baseline models in terms of fidelity and sampling speed. As we scale RVQ depth, our model exhibits improvements in generation fidelity or efficiency, showing its scalability and generalizability across different modalities.

---

[1]https://resgen-ai.github.io/

## Acknowledgement

The authors would like to express our gratitude to Kangwook Lee for the valuable discussions. We also extend our thanks to Beomsoo Kim, Seungjun Chung, Dong Won Kim, and Gibum Seo for their essential support in data handling and verification. This research (paper) used datasets from 'The Open AI Dataset Project (AI-Hub, S. Korea)'. All data information can be accessed through 'AI-Hub (www.aihub.or.kr)'.

## Impact Statement

Our work advances the field of generative modeling by introducing a memory-efficient approach for high-fidelity sample generation using Residual Vector Quantization (RVQ). The proposed model, ResGen, demonstrates significant improvements in generation quality and efficiency across challenging modalities such as image generation and text-to-speech synthesis. As data resolution and size continue to increase in real-world applications, our method provides a scalable and efficient solution, reducing the memory and computational burden typically associated with high-resolution generative tasks.

The potential societal benefits of this research are substantial, particularly in areas where efficient and high-quality generation is critical, such as accessibility technologies, creative industries, and scientific simulations. For example, ResGen can facilitate resource-efficient generation of synthetic media, enabling small-scale researchers and organizations to access and utilize state-of-the-art generative technologies that were previously out of reach.

However, as with all generative models, there are potential misuse risks, such as the creation of synthetic media for deceptive purposes or the unauthorized replication of copyrighted content. To mitigate these risks, we encourage researchers and practitioners to adopt responsible use policies, including mechanisms to detect and authenticate synthetic content, and to promote transparency in model development and deployment.

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

# A. Training details

## A.1. Configurations for Training

**Vision models**   We train our method using an architecture similar to DiT (Peebles & Xie, 2023), adopting the XLarge version while modifying the adaptive layer normalization layers for conditioning by replacing their linear layers with bias parameters. As shown in Table 1, all generative models are trained for 2.8M iterations under the same RVQ token setting. The model sizes are as follows: ResGen (576M), AR-ResGen (625M), RQ-Transformer (626M), and MaskGIT (580M). To maintain fair and consistent evaluation conditions across models, RQ-Transformer is assessed in this experiment using classifier-free guidance (CFG), rather than the rejection sampling adopted in its original publication.

In Table 2, all variants of ResGen are trained with a batch size of 256 across 4 GPUs for 7M iterations. The masking scheduling function $\gamma(\cdot)$ is defined as $\gamma(r) = (1 - r^2)^{\frac{1}{2}}$ and applied throughout all training.

To increase the depth of RVQ, we warm-start from the 4-depth RQ-VAE checkpoint (Lee et al., 2022), excluding the attention layers, and reduce the latent dimension from 256 to 64. Each VAE is further trained for an additional 1M steps, both with and without adversarial training, following the same configuration as prior work. For the RVQ quantizer, we employ the probabilistic RVQ method from Kim et al. (2024), which updates the RVQ codebook embeddings proportionally based on how closely the VAE latents match each embedding. The codebook size at each depth is set to 1024. The resulting token embeddings obtained from the RVQ quantizer are then fed into ResGen, where they are projected to match the hidden size via a linear layer.

**Audio models**   For the Text-to-Speech task, our model, based on the DiT XLarge architecture as in the vision task, is trained using the same configuration as in prior work (Kim et al., 2024), utilizing 4 GPUs for 310M iterations. The masking scheduling function $\gamma(\cdot)$ is defined as $\gamma(r) = (1 - r^2)^{\frac{1}{2}}$ and applied throughout all training. We employ 4 transformer layers to train a linear regression duration predictor for the text inputs, built on top of the pretrained text encoder, ByT5-Large (Xue et al., 2022). The duration predictor is trained to minimize the L2 loss between the mel-spectrogram and the expanded hidden representation of text from ByT5, with alignment achieved using the monotonic alignment search algorithm (Kim et al., 2020). The expanded text hidden representation is downsampled through a strided convolution layer to match the length of the RVQ token sequence and combined with the embeddings of RVQ tokens obtained from the RVQ quantizer. These combined representations are then projected to match the model's hidden size using a linear layer.

To increase the depth of RVQ from 32 in MelVAE to 72, we train a separate autoencoder that directly processes waveforms. The 44 kHz waveforms are transformed using the Short-Time Fourier Transform (STFT) with a hop length of 8 and a window size of 32. The real and imaginary parts are concatenated and then encoded through an encoder composed of three blocks, each consisting of three 1D ConvNeXt layers (Siuzdak, 2024; Liu et al., 2022) followed by a strided convolution layer with a stride of 8 and a kernel size of 8. This results in a final latent representation of 512 dimensions with a temporal resolution of 10.7 Hz, matching that of MelVAE. The decoder mirrors the encoder symmetrically, reconstructing STFT parameters, which are then converted back to waveforms via inverse STFT. The autoencoder is trained with similar training configurations as DAC, including mel-spectrogram reconstruction loss and adversarial losses. The RVQ codebook size at each depth is set to 1024, and the entire autoencoder model consists of 185 million parameters.

## A.2. Implementation Techniques for Training

**Mixture of Gaussians Implementation.**   Our model utilizes a mixture of Gaussian distributions to represent the distribution over latent embeddings. Specifically, for each token position $i$, the model outputs the mixture probabilities $\boldsymbol{\pi}_i = \{\pi_i^{(\nu)}\}_{\nu=1}^{K}$, the mean vectors for each mixture component $\{\boldsymbol{\mu}_i^{(\nu)}\}_{\nu=1}^{K}$, and additional scale and shift parameters for affine transformations $a_i \in \mathbb{R}$ and $\boldsymbol{b}_i \in \mathbb{R}^H$, where $K$ is the number of mixture components and $H$ is the embedding dimension. When projecting the hidden output (size $O$) to $K$ mixture probabilities, it leads to a projection complexity dominated by $\mathcal{O}(O * K * H)$.

For our vision model ($O = 1152, K = 1024, H = 64$), this cost is comparable to a standard softmax layer with a ~64K vocabulary ($V = K * H$), which is practical. For higher-dimensional embeddings like in audio ($H = 512$), we use low-rank projection for the means, reducing the dominant computational term to $\mathcal{O}(O * K * h + H * h)$, again making the effective cost similar to a ~64K vocabulary model ($h = 64$). This technique, previously used in CLaM-TTS(Kim et al., 2024), significantly mitigates overhead.

**Training Objective Modification.** From Equation 1, the log-likelihood of the target embedding $z_i$ is formulated as $\log p_\theta(z_i | x \odot m) = -\log a_i + \log \sum_\nu \pi_i^{(\nu)} \mathcal{N}(\tilde{z}_i; \mu_i^{(\nu)}, I)$, where $\tilde{z}_i = (z_i - b_i)/a_i$. To further encourage the usage of every mixture component, we modify the objective by decomposing it into a sum of classification and regression losses. Similar to prior work (Kim et al., 2024), applying Jensen's inequality, we have:

$$-\log a_i - \log \sum_\nu \pi_i^{(\nu)} \mathcal{N}(\tilde{z}_i; \mu_i^{(\nu)}, I)$$

$$\leq -\log a_i \underbrace{- \sum_\nu q(\nu \mid \tilde{z}_i, \mu_i) \log \mathcal{N}(\tilde{z}_i; \mu_i^{(\nu)}, I)}_{\text{regression loss}} + \underbrace{D_{\text{KL}}(q(\nu \mid \tilde{z}_i, \mu_i) \parallel \pi_i)}_{\text{classification loss}},$$

where $q(\nu \mid \tilde{z}_i, \mu_i)$ is an auxiliary distribution defined as $q(\nu \mid \tilde{z}_i, \mu_i) \propto \mathcal{N}(\tilde{z}_i; \mu_i^{(\nu)}, I)$. This choice of $q$ ensures that mixture components with mean vectors closer to $\tilde{z}_i$ have higher probabilities, while all components retain non-zero probabilities. Consequently, every mixture component contributes to the training process, promoting higher component usage and diversity in the model's predictions.

**Low-rank Projection.** Increasing the number of mixture components $K$ leads to a substantial growth in the output dimensionality of the model, as it scales with $K \times H$. To accommodate a high number of mixtures without incurring excessive computational costs, we adopt a low-rank projection approach following the methodology of the prior work (Kim et al., 2024).

In this approach, the model outputs low-rank mean vectors $\{\tilde{\mu}_i^{(\nu)}\}_{\nu=1}^K$, which are then transformed using trainable parameters $M^{(\nu)}$ and $s^{(\nu)}$: $\mu_i^{(\nu)} = M^{(\nu)} \tilde{\mu}_i^{(\nu)} + s^{(\nu)}$. This decomposition allows for efficient computation of the squared distance $\|\tilde{z}_i - \mu_i^{(\nu)}\|^2$ by expanding it as follows:

$$\|\tilde{z}_i - \mu_i\|^2 = \|\tilde{z}_i - (M\tilde{\mu}_i + s)\|^2$$
$$= \tilde{z}_i^T \tilde{z}_i + \tilde{\mu}_i^T (M^T M)\tilde{\mu}_i + s^T s - 2(M^T \tilde{z}_i)^T \tilde{\mu}_i - 2\tilde{z}_i^T s + 2\tilde{\mu}_i^T M^T s, \tag{3}$$

where we omit $\nu$ for simplicity. This low-rank projection enables the model to handle a large number of mixture components without significant overhead, thereby enhancing both the scalability and performance of the generative process.

## A.3. Pseudo-code for Training

---

**Algorithm 1** Training

---

1: **procedure** BinaryMask($n, L, D$)
2:     Sample $\boldsymbol{k}_{1:L}$ **without replacement** with total draws $n$.
3:     **for** $i = 1$ to $L$ **do**
4:         $\boldsymbol{m}_{i,1:(D-k_i)} \leftarrow 1$
5:         $\boldsymbol{m}_{i,(D-k_i+1):D} \leftarrow 0$
6:     **end for**
7:     **return** $m$
8: **end procedure**
9:
10: **repeat**
11:     $\boldsymbol{x} \sim p_{data}$
12:     $r \sim \text{Uniform}[0, 1]$
13:     $n \leftarrow \lceil \gamma(r) \cdot L \cdot D \rceil$
14:     $\boldsymbol{m} \leftarrow \text{BinaryMask}(n, L, D)$
15:     $\boldsymbol{z} \leftarrow \sum_j \left( \boldsymbol{e}(\boldsymbol{x}_{:,j}; j) \odot (1 - \boldsymbol{m}_{:,j}) \right)$
16:     Take a gradient descent step on:
17:         $\nabla_\theta \mathcal{L}_{\text{mask}}(\boldsymbol{x}, \boldsymbol{m}; \theta)$
18: **until** converged

---

## A.4. Scalability of our generative modeling with RVQ depth

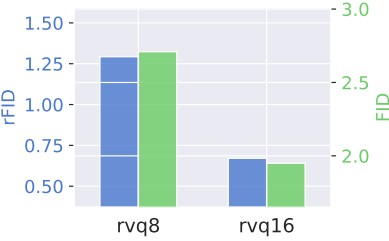

*Figure 3.* Effect of RVQ depth on both the autoencoder's reconstruction quality and our method's generation quality. We compare two configurations, *rvq8* and rvq16, corresponding to RVQ depths of 8 and 16, respectively. As the RVQ depth increases, the autoencoder achieves better reconstruction quality (lower rFID). Despite the increased number of tokens at deeper depth, our generative models show better generation quality (lower FID), underscoring its scalability with deeper quantization.

# B. Sampling details

## B.1. Sampling with Confidence Scores

Inspired by confidence-based sampling with a choice temperature, as proposed in MaskGIT (Chang et al., 2022) and GIVT (Tschannen et al., 2024), we unmask tokens based on the log probabilities computed for all masked tokens. These log probabilities are derived from the squared distance between token embeddings and the sampled latent $\boldsymbol{z}_i$ at each position $i$. The log probability $\log p(x_{i,j} \mid \boldsymbol{z}_i)$ is calculated as $\log p(x_{i,j} \mid \boldsymbol{z}_i) \propto \log \mathcal{N} \left( \boldsymbol{z}_i - \sum_{d=1}^{j-1} \boldsymbol{e}(x_{i,d}; d); \boldsymbol{e}(x_{i,j}; j), \sigma_j^2 I \right)$ for all masked positions $i$ and $j$. Here, $\sigma_j$ denotes the standard deviation of latents at RVQ depth $j$ pre-calculated during RVQ training. The log probabilities are cumulatively summed across depths, and the confidence score is obtained by adding

Gumbel noise, scaled by the choice temperature, to them. The choice temperature remains fixed throughout all inference steps in our settings. Tokens with higher confidence scores are prioritized for unmasking and are filled earlier in the iterative generation process. In particular, Table 4 lists the exact hyper-parameters used during sampling.

*Table 4.* Sampling hyper-parameters for the experiments in Figure 2. The *cfg schedule* indicates a linear scaling of the classifier-free-guidance weight from the *start* value at the first step to the *end* value at the last step. *Top-p* is the nucleus-sampling threshold and $\tau$ is the temperature.

| Variant | Steps | cfg schedule | Top-p | $\tau$ |
|---|---|---|---|---|
| ResGen-rvq16 | 28 | $0.02 \to 2.4$ | 0.94 | 28.0 |
| | 48 | $0.02 \to 2.4$ | 0.96 | 28.0 |
| | 64 | $0.02 \to 2.2$ | 0.98 | 28.0 |
| ResGen-rvq8 | 28 | $0.02 \to 2.4$ | 0.94 | 28.0 |
| | 48 | $0.02 \to 2.4$ | 0.96 | 28.0 |
| | 64 | $0.02 \to 2.2$ | 0.98 | 28.0 |

## B.2. Ablation Studies on Sampling

We conducted ablation experiments to analyze the characteristics of our sampling algorithm, focusing on hyperparameters such as sampling steps, top-p values, and temperature scale, and their impact on generation quality. As illustrated in Figure 4a, increasing the number of sampling steps improves generation quality in both scenarios: with classifier-free guidance (CFG) and without it. This demonstrates that additional steps enable the model to refine its outputs more effectively, resulting in higher-quality generations.

Figure 4b explores the impact of top-p values on generation quality, highlighting different trends depending on the use of CFG. With CFG, higher top-p values promote greater diversity in sampling, resulting in improved generation quality. In contrast, without CFG, lower top-p values lead to reliance on the model's confident predictions, thereby enhancing generation quality. These findings suggest that adjusting the top-p value can be beneficial, particularly in the absence of CFG.

Finally, Figure 4c highlights the influence of temperature on generation quality. A moderate temperature introduces controlled stochasticity during sampling, mitigating the monotonicity inherent in RVQ-token unmasking, where the order is guided by confidence scores. By adjusting the temperature, we achieve a balance between diversity and fidelity, optimizing overall generation performance.

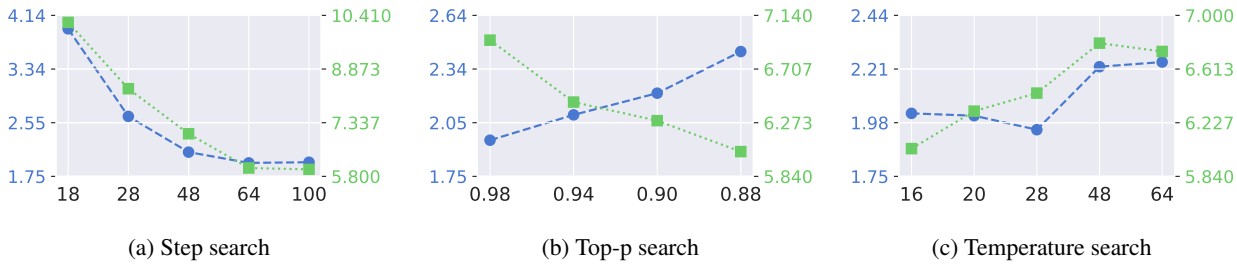

(a) Step search    (b) Top-p search    (c) Temperature search

*Figure 4.* Configuration search results for sampling methods with (blue) and without (green) classifier-free guidance (CFG). (a) The effect of varying the number of sampling steps, (b) the impact of different top-p values, and (c) the influence of temperature scaling on confidence scores.

## B.3. Pseudo-code for Sampling

---

**Algorithm 2** Sampling

---

1: **procedure** BinaryUnmask($n, L, D, \boldsymbol{m}$)
2:     Compute the number of masked tokens $q_i = \sum_{j=1}^{D}(1 - m_{i,j})$
3:     Sample $\boldsymbol{k}_{1:L}$ from a multivariate hypergeometric distribution with maximum number of selection $q_i$, total draws $\sum_i q_i - n$.
4:     **for** $i = 1$ to $L$ **do**
5:         $\boldsymbol{m}[i, (D - q_i + 1){:}(D - q_i + k_i)] \leftarrow 1$
6:     **end for**
7:     **return** $\boldsymbol{m}$
8: **end procedure**
9:
10: Initialize a fully masked sequence $\boldsymbol{x} \in \mathbb{N}^{L \times D}$
11: Initialize mask $\boldsymbol{m} \in \{0, 1\}^{L \times D}$ with zeros.
12: **for** $t = 1, \ldots, T$ **do**
13:     $\boldsymbol{z} \sim p_\theta(\boldsymbol{z}|\boldsymbol{x} \odot \boldsymbol{m})$
14:     Apply residual vector quantization for masked tokens:
15:         $\boldsymbol{x} \leftarrow RVQ(\boldsymbol{z}, \boldsymbol{m})$
16:     $r \leftarrow \frac{t}{T}$
17:     $n \leftarrow \lceil \gamma(r) \times L \times D \rceil$
18:     **if** using random sampling **then**
19:         $\boldsymbol{m} \leftarrow$ BinaryUnmask($n, L, D, \boldsymbol{m}$)
20:     **else if** using confidence-based sampling **then**
21:         Update $\boldsymbol{m}$ by selecting $n$ tokens with the confidence-based sampling from Section B.1.
22:     **end if**
23: **end for**
24: Return $\boldsymbol{x}$

---

# C. Additional Analysis

## C.1. Elaborating Differences with VAR

While both ResGen and VAR leverage the hierarchical structure of residual vector quantization (RVQ), they differ in how this hierarchy is applied throughout the model, leading to key distinctions in representation, generation, and adaptability.

In terms of structure and resolution, VAR assigns each RVQ depth to a distinct spatial grid (e.g., 1×1, 2×2, up to the original resolution), requiring a fixed depth–resolution hierarchy to be specified in advance. In contrast, ResGen applies all quantization depths to refine a single latent grid whose length matches the VAE encoder's output sequence. This difference also induces the generative process. VAR generates tokens sequentially across depths, meaning that depth $k$ can only be generated after completing depth $k - 1$, although parallel sampling is possible within each depth. ResGen, on the other hand, predicts cumulative embeddings that aggregate information from multiple depths in a single masked-generation step, enabling fully parallel prediction along the sequence-length dimension.

Furthermore, ResGen offers greater flexibility for modeling data across diverse domains. Because VAR's design depends on a predefined resolution schedule, adapting it to domains with variable or arbitrary output lengths—such as text-to-speech—can be non-trivial. In contrast, ResGen employs a length-agnostic tokenization scheme, enabling straightforward application to such domains without requiring changes to the model structure. Finally, in terms of resolution flexibility, increasing the quantization depth in RVQ (e.g., to 16) allows ResGen to represent information at lower effective spatial resolutions (e.g.,

$8\times 8$) without extending the sequence length. Achieving comparable resolution is less direct in VAR's depth-resolution coupled structure, and it would require redefining the entire depth–resolution mapping.

## C.2. Further comparison with faster MAR

To ensure a fair comparison with MAR and to directly address the question of how ResGen performs compared to faster MAR variants, it is important to clarify a key difference in how the generation speed is reported. MAR evaluates efficiency using throughput (i.e., seconds per image averaged over a large batch), whereas we report wall-clock time for generating a single image on one A100 GPU. This discrepancy in measurement protocols largely explains the apparent difference in reported latency.

Beyond the reporting protocols, fundamental differences in the generative process also distinguish ResGen from MAR. MAR utilizes multi-token prediction followed by continuous diffusion steps, a process that allows for iterative refinement of continuous-valued tokens. In contrast, ResGen performs disjoint token unmasking at each step without revising tokens decided in prior steps. This architectural divergence contributes to differing performance characteristics, particularly highlighted by the impact of classifier-free guidance (CFG). Furthermore, although MAR-B can perform well without CFG, achieving its best results, especially with guidance, often requires significantly more sampling steps (e.g., 100+ diffusion steps) compared to ResGen's typically more constrained number of effective steps. This presents a trade-off between MAR-B's parameter efficiency and ResGen's inference efficiency, particularly when aiming for high-quality results with guidance.

To evaluate whether ResGen still outperforms speed-optimized MAR variants under these varied conditions and considering these modeling differences, we conducted additional experiments in controlled settings. The first set of experiments reduced the number of auto-regressive (AR) steps in MAR while keeping the diffusion step count fixed at 100. As shown in Table 5, even with as few as 16 steps - the fastest configuration we tested - MAR-B achieved an FID of 4.11, substantially worse than ResGen -rvq16 (FID 1.93), even if this MAR-B configuration took roughly 5.6 seconds per image.

Additionally, we reduced the number of diffusion steps in MAR (with AR steps fixed at 256). As shown in Table 6, MAR-B with only 25 diffusion steps required approximately 22.5 seconds per image, yet still underperformed relative to ResGen, with an FID of 3.38 compared to 1.93. These results highlight that ResGen consistently achieves better sample quality at significantly lower latency, even when compared to aggressively optimized MAR configurations.

Table 5. FID ($\downarrow$) for MAR-B/L when varying AR steps with fixed diffusion steps.

| Model | AR = 64 | AR = 32 | AR = 16 |
|---|---|---|---|
| MAR-B | 2.33 | 2.44 | 4.11 |
| MAR-L | 1.81 | 2.10 | 4.32 |

Table 6. FID ($\downarrow$) for MAR-B/L when varying diffusion steps with fixed AR steps.

| Model | Diff = 100 | Diff = 50 | Diff = 25 |
|---|---|---|---|
| MAR-B | 2.31 | 2.39 | 3.38 |
| MAR-L | 1.78 | 1.83 | 2.22 |

## C.3. Sensitivity to Masking Strategies

We investigated the sensitivity of ResGen to different masking schedules during training and sampling. Specifically, we trained ResGen-rvq16 models (400K iterations) using three distinct masking strategies: *cosine*, *circle*, and *exponential*. At sampling time, each model was evaluated both under its original training schedule and under a different one, resulting in a cross-evaluation of masking strategies. All evaluations were conducted with and without classifier-free guidance (CFG), using exponential moving average (EMA) checkpoints.

Interestingly, as shown in Table 7, the best performance (FID 9.53 at 64 steps with CFG) was achieved when training with the *circle* masking strategy but sampling with the *exponential* strategy. This suggests that the choice of masking schedule at inference time can have a meaningful impact, even when it differs from the training configuration. The exponential

schedule, in particular, unmasks fewer tokens in the early stages and progressively increases the unmasking rate, creating a coarse-to-fine decoding process. This appears to benefit ResGen by allowing it to first establish a stable global structure before refining finer details. Notably, exponential sampling yielded strong performance even when the model was not trained with it, highlighting its robustness and potential as a generally effective inference schedule for masked generation with RVQ tokens.

*Table 7.* FID scores for ResGen-rvq16 under different combinations of training and sampling masking strategies. Sampling 64, 48, and 28 steps, with and without classifier-free guidance (CFG).

| CFG | Training | Sampling | 64 steps | 48 steps | 28 steps |
|-----|----------|----------|----------|----------|----------|
| w/o | cosine | cosine | 28.13 | 28.01 | 29.18 |
| w/o | cosine | exp | 32.30 | 32.70 | 32.81 |
| w/o | circle | circle | 26.04 | 26.41 | 26.73 |
| w/o | circle | exp | 32.44 | 33.12 | 33.92 |
| w/o | exp | exp | 41.12 | 41.67 | 41.87 |
| w/ | cosine | cosine | 15.46 | 15.69 | 17.27 |
| w/ | cosine | exp | 9.66 | 9.78 | 10.08 |
| w/ | circle | circle | 10.09 | 10.35 | 10.44 |
| w/ | circle | exp | **9.53** | 9.72 | 9.98 |
| w/ | exp | exp | 12.63 | 12.65 | 12.83 |

*Table 8.* Performance of ResGen-rvq16 compared against leading masked generative models, MAGVIT-v2 and MaskBiT, on image generation. Boldface denotes the best FID score; an asterisk (*) marks scores reported in their original papers.

| Model | Code length | FID (w/ CFG) ↓ | Inference Steps |
|-------|-------------|----------------|-----------------|
| ResGen-rvq16 | 64 | 1.93 | 63 |
| MAGVIT-v2 | 256 | 1.78* | 64 |
| MaskBiT | 256 | **1.62*** | 64 |

## C.4. Comparison with Recent Masked Generative Models

We analyze ResGen's performance against recent leading masked generative models, MAGVIT-v2 (Yu et al., 2024) and MaskBit (Weber et al.). As shown in Table 8, these models achieve FID scores of 1.78 and 1.62, respectively. Our ResGen-rvq16 (with CFG) achieves an FID of 1.93 with a comparable number of sampling steps. While their FID scores are lower than 1.93 of our ResGen-rvq16, ResGen presents distinct advantages stemming from its architectural design and modeling approach:

**Modeling Cross-Depth Token Dependencies.** ResGen predicts cumulative embeddings, thereby explicitly modeling dependencies across the RVQ depth dimension. This contrasts with approaches like MAGVIT-v2 and MaskBit, which predict independent groups of bits from Lookup-Free Quantization (LFQ). Notably, MaskBit's own ablation study (Table 3b in their work (Weber et al.)) indicates that performance can degrade as the number of independent bit groups increases larger than two, suggesting potential challenges in scaling to very high fidelity (which would necessitate more bits or groups). ResGen's strategy of directly modeling these correlations may offer enhanced scalability, particularly for deep quantization schemes. The benefit of this explicit correlation modeling is further supported by an internal ablation of ResGen (see Section 5.2.1 and Table 1, left): a variant predicting discrete tokens directly in parallel across all depths (instead of cumulative embeddings) demonstrates strong performance, but slightly underperforms compared to our final ResGen using cumulative embeddings, underscoring the efficacy of our chosen strategy.

**Resolution and Depth Flexibility.** ResGen's proficiency in handling deep RVQ (e.g., 16-depth) facilitates the use of a lower spatial resolution for the token map (e.g., $8 \times 8$), while maintaining high reconstruction quality, compared to typical VQ-based methods that use $16 \times 16$ spatial tokens. The efficient RVQ depth handling in ResGen makes this trade-off viable, offering valuable flexibility in balancing representational capacity and memory footprint.

These distinctions highlight ResGen's unique strengths in managing complex token dependencies and architectural flexibility.

## D. Limitations and Future Directions

The proposed method demonstrates favorable memory efficiency alongside competitive sampling speed and generation quality. One potential avenue for further improvement that is not explored in our work is the utilization of key-value (KV) caching within the transformer architecture. By progressively filling tokens, positions that have been completely filled can reuse their precomputed KV values, thereby reducing redundant computations. This strategy can significantly enhance the sampling speed and reduce overall computational overhead, making it a promising direction for future research.

While our approach is intricately designed around Residual Vector Quantization (RVQ) tokens, recent developments in quantization methods suggest that Finite Scalar Quantization (FSQ) may offer additional benefits (Mentzer et al., 2024). Extending our approach to support FSQ, however, is not straightforward, as it involves distinct tokenization and embedding processes. Nevertheless, exploring this direction could lead to novel quantization strategies and improved generative performance.

Another key observation is that our approach achieves high-quality generation with a relatively small number of iterations. We hypothesize that this efficiency, compared to conventional diffusion models, stems from the unmasking process rather than a denoising process. Since predicting tokens based on completely unmasked tokens is likely easier than predicting them based on noisy inputs, the model benefits from the simpler prediction task. Despite this empirical success, our work lacks a theoretical justification for why such a low number of inference steps is sufficient. Providing a formal theoretical and analytic explanation for this phenomenon represents another promising future direction.

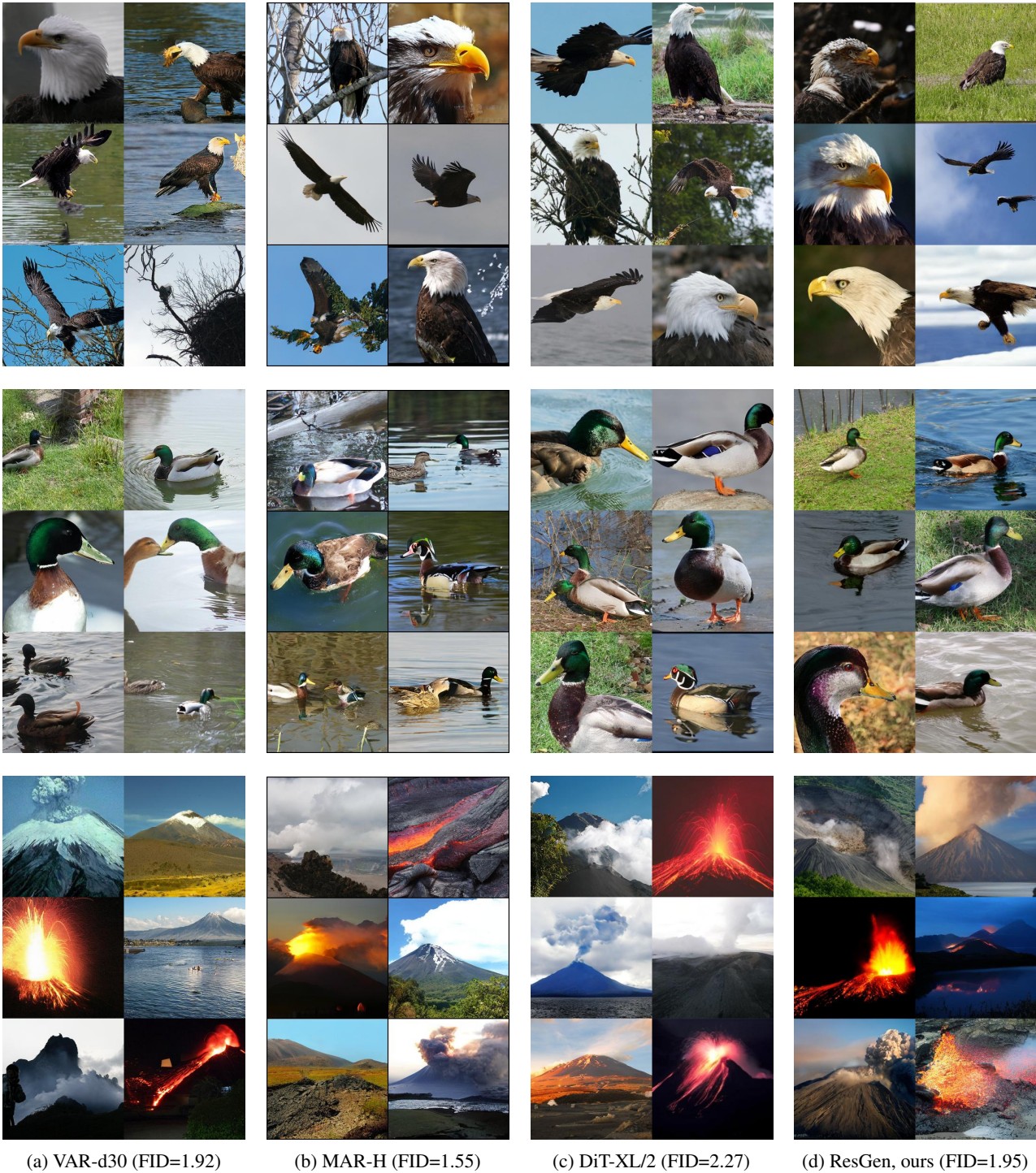

(a) VAR-d30 (FID=1.92)   (b) MAR-H (FID=1.55)   (c) DiT-XL/2 (FID=2.27)   (d) ResGen, ours (FID=1.95)

*Figure 5.* Model comparison on ImageNet 256×256 benchmark.

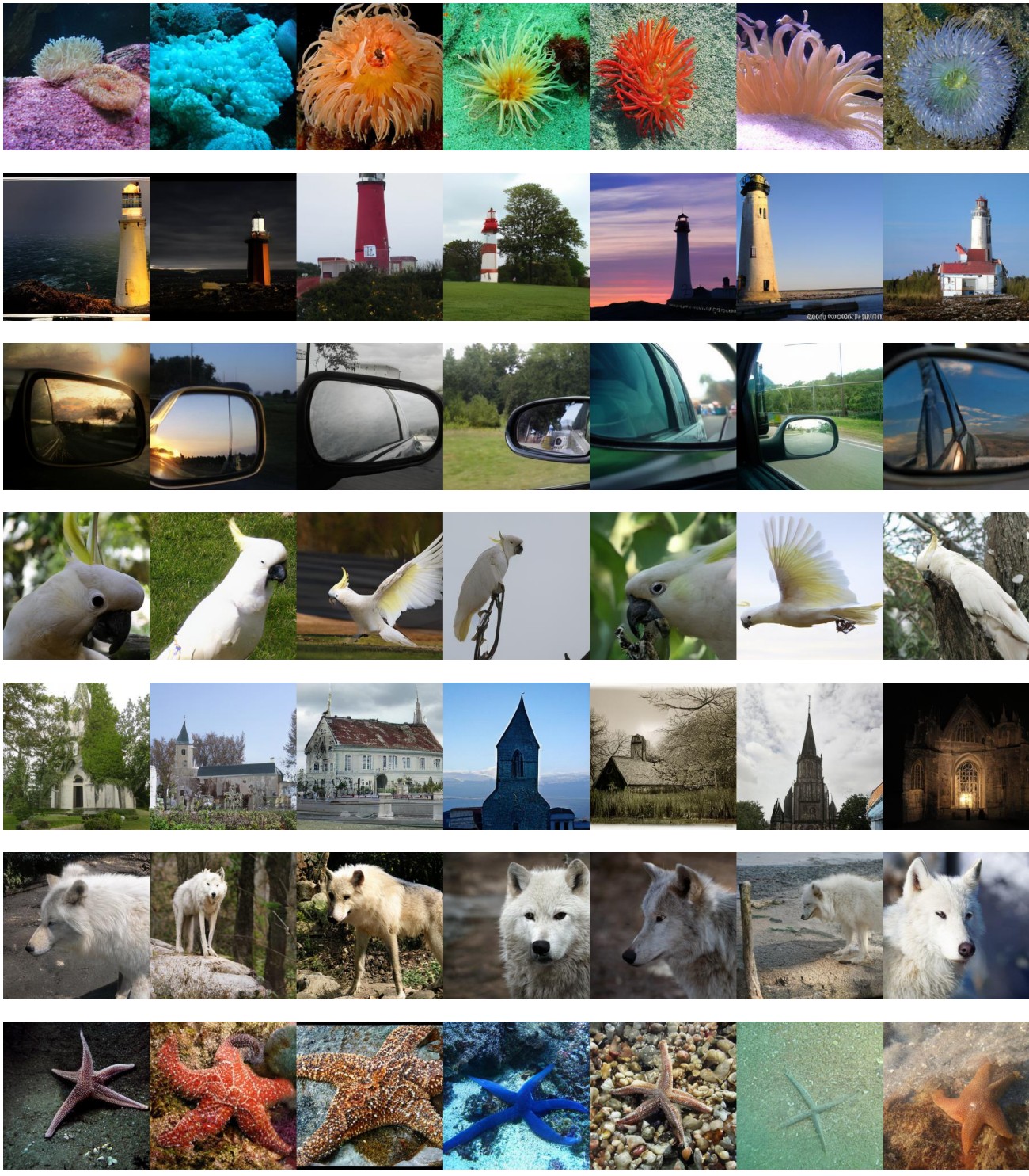

*Figure 6.* Randomly generated 256×256 samples by ResGen trained on ImageNet.

