# OpenReview forum: "Efficient Generative Modeling with Residual Vector Quantization-Based Tokens"
_ICML.cc/2025/Conference — ICML 2025 poster_

### Official Review · Reviewer_aQEq · 2025-03-08

**Overall Recommendation:** 3

**Summary:**

This paper proposes an efficient generative framework (ResGen) to model residual tokens that has an additional depth dimension in the code sequences. Specifically, ResGen adopts the masked generative framework to sample tokens, and leverages Gaussian mixtures to directly predict the sum of masked token embeddings. This method achieves strong results with improved efficiency on both class-conditional image generation and text-to-speech tasks.

## update after rebuttal
The authors' response addressed my concerns regarding the efficiency of the proposed GMM head, in comparison with the AR head in RQ-Transformer. Therefore, I have adjusted my rating to borderline accept. However, I observed that the major performance gains in FID primarily come from the use of a MaskGIT-like generation framework, which is not an innovation of this work. I recommend that the authors clarify this point in their paper to ensure transparency and accuracy in reporting their contributions.

**Claims And Evidence:**

The claims in this paper are supported by experimental results or prior studies.

**Essential References Not Discussed:**

N/A

**Experimental Designs Or Analyses:**

The experimental designs are sound.

**Methods And Evaluation Criteria:**

I am not familiar with text-to-speech evaluation. The metrics used for class-conditional image generation are common and standard.

**Other Comments Or Suggestions:**

Please see weaknesses.

**Other Strengths And Weaknesses:**

**Strengths:**
- The paper is well written and easy to follow.
- The proposed method can be generally applied to both image generation and text-to-speech generation.

**Weaknesses:**
- Residual tokens have both length and depth dimensions. The focus of ResGen is to model **the depth dimension** more efficiently, i.e., by predicting embeddings of depth tokens collectively rather than individually. However, it is not clear to what extent does this method improve the efficiency over its baseline, i.e., a depth transformer in RQ-Transformer. Notably, ResGen adopts the masked generative framework to model **the length dimension**, which requires much fewer sampling steps than the autoregressive framework used in RQ-Transformer. This makes ResGen and RQ-Transformer less comparable. The authors are encouraged to provide a strict ablation on this. For example, using the the same autoregressive framework to model the length dimension.

- Comparisons with recent masked generative models such as MAGVIT-v2 [1] and MaskBit [2] are missing in Table 1 and Figure 2. It seems ResGen does not show clear advantages in terms of both generation quality and sampling speed over these methods.

[1] Yu, Lijun, et al. "Language Model Beats Diffusion-Tokenizer is key to visual generation." The Twelfth International Conference on Learning Representations.

[2] Weber, Mark, et al. "MaskBit: Embedding-free Image Generation via Bit Tokens." Transactions on Machine Learning Research.

**Questions For Authors:**

1. Why not include VAR-d30 in Figure 2 for wallclock time comparison?

2. Could the authors provide the exact coordinates of the data points in Figure 2 (left one)?

**Relation To Broader Scientific Literature:**

The key contribution of this paper is to model residual tokens more efficiently.

**Theoretical Claims:**

I did not fully check the details of the theoretical proofs in Section 3.2.

---

> ### Author Rebuttal · Authors · 2025-04-01
>
> We sincerely thank the reviewer for the constructive feedback. We address the specific weaknesses and questions raised below:
>
> **Regarding W1: Isolating the Efficiency Gains of Depth Modeling**
> We agree that comparing ResGen (masked generation) directly with RQ-Transformer (autoregressive generation) makes it difficult to isolate the efficiency gains specifically from our depth modeling strategy versus the gains from the masked generation framework itself. To provide a stricter ablation, as suggested:
> *   **New Experiment:** We created an "AR-ResGen" variant. This model uses the *exact same spatial autoregressive transformer* as the RQ-Transformer baseline. The key difference is that it replaces RQ-Transformer's autoregressive depth transformer with our *cumulative embedding prediction MLP* (of similar size) to handle the depth dimension. This isolates the effect of the depth modeling approach while keeping the sequence modeling framework constant (autoregressive).
> *   **Results:** This AR-ResGen achieves better FID scores than the original RQ-Transformer, particularly with few iterative refinement steps for the depth prediction, demonstrating the efficiency of our cumulative embedding approach even within an AR framework. The results (FID) are:
>     | AR-ResGen Iterations | w/o CFG | w/ CFG |
>     | :------------------- | :------ | :----- |
>     | 1                    | 27.45   | 6.47   |
>     | 2                    | 24.10   | 5.33   |
>     | 4                    | 23.75   | 5.30   |
>     | 8                    | 23.48   | 5.22   |
> *   **Conclusion:** This targeted ablation confirms that our cumulative embedding prediction for depth modeling is inherently more efficient and effective than standard autoregressive depth handling, independent of the sequence generation strategy. We will include these findings in the revised manuscript.
>
> **Regarding W2: Comparison with MAGVIT-v2 and MaskBit**
> We thank the reviewer for highlighting the missing comparisons with MAGVIT-v2 and MaskBit. We acknowledge their strong performance (FID 1.78 and 1.62) and will add these results to Table 1 and Figure 2 for context. Our current FID is 1.93 with similar steps. While their FID scores are currently lower, ResGen offers distinct advantages:
> *   **Theoretical Advantage (Modeling Correlations):** ResGen predicts cumulative embeddings to explicitly model dependencies across the RVQ depth dimension. This differs from methods like MAGVIT-v2/MaskBit using Lookup-Free Quantization (LFQ) and predicting independent bit groups. MaskBit's own ablation (Table 3b) shows performance degrading as independent groups increase, suggesting potential scaling limitations for higher fidelity (requiring more bits/groups). ResGen's approach of modeling correlations directly may offer better scalability for deep quantization or numerous discrete outputs. To show the benifit of modeling these correlation between tokens across depth, we implemented a variant of our method which uses the same masked generation framework introduced in our method section but predicts discrete tokens directly and in parallel across all depths at each step, instead of predicting continuous cumulative embeddings. This variant achieved FID scores of 12.79 (w/o CFG) and 2.91 (w/ CFG). While this performance surpasses the other comparison methods evaluated in our study, it is slightly worse than our final proposed model which predicts cumulative embeddings, thus supporting the efficacy of the cumulative embedding strategy to capture such token correlations.
> *   **Practical Advantage (Resolution/Depth Trade-off):** Our use of 16-depth RVQ enables a lower spatial resolution (8x8) than typical 16x16 VQ methods, while achieving high reconstruction quality measure in rFID. ResGen's efficient depth handling makes this viable, offering flexibility in memory usage and model design.
> We will clarify these points and add the comparisons to the manuscript.
>
> **Regarding Q1: Inclusion of VAR-d30 in Figure 2**
> Thank you for this suggestion. To provide a more complete comparison of performance and efficiency trade-offs, we agree that including VAR-d30 in the wallclock time comparison (Figure 2) would be beneficial. We will update Figure 2 to include VAR-d30 in the revised manuscript.
>
> **Regarding Q2: Exact Coordinates for Figure 2 (left)**
> Certainly. The exact coordinates (Wallclock Time [s], FID) for the data points in Figure 2 (left) are:
> *   DiT: [ (4.68, 2.27) ]
> *   VAR: [ (0.16, 3.60), (0.19, 2.95), (0.24, 2.33) ]
> *   MAR: [ (89.69, 2.31), (105.6, 1.78), (133.01, 1.55) ]
> *   RQTran: [ (3.61, 3.89), (3.73, 3.80) ]
> *   ResGen-rvq8: [ (0.84, 2.87), (1.43, 2.78), (1.86, 2.75) ]
> *   ResGen-rvq16: [ (0.98, 2.26), (1.67, 2.14), (2.28, 1.98) ]
> *   MaskGiT: [ (0.98, 6.18) ]
> Solid lines connect different models; dashed lines connect points for the same model with varying sampling steps. We will add this detailed information, likely in the figure caption or appendix, for clarity in the revision.

---

> > ### Comment · Reviewer_aQEq · 2025-04-02
> >
> > Thanks for the authors' response, which addresses some of my concerns. However, I see that the gFID of "AR-ResGen" is not notably better than that of RQ-Transformer (5.50 as shown in Table 1 of the paper). Meanwhile, while AR-ResGen requires 2~4x fewer sampling steps in the depth dimension, this reduction does not appear to result in significant speed improvements, given that the original depth transformer of the RQ-Transformer is lightweight and introduces minimal computational overhead. To fully convince me, the authors could provide the wallclock time and FID for AR-ResGen as presented in Figure 2.
> >
> > Besides, according to the rebuttal, the FID of VAR does not match the numbers presented in Table 2 of the paper. Please explain the reasons. It seems VAR to be a more efficient choice than ResGen as it is roughly 10 times faster than ResGen and achieves similar FID (e.g., VAR (0.19, 2.95) v.s., ResGen-rvq8 (1.86, 2.75)).

---

> > > ### Author Response · Authors · 2025-04-02
> > >
> > > We appreciate the reviewer’s thoughtful follow-up and agree that practical efficiency, especially in terms of wallclock time, is essential to clearly demonstrate the advantages of our cumulative embedding prediction strategy.
> > >
> > > **Concerning the efficiency comparison between RQ-Transformer and AR-ResGen:** To directly address this concern, we conducted additional experiments measuring single-sample generation time using an NVIDIA A100 GPU, comparing the original RQ-Transformer with our AR-ResGen variant. The results are summarized below:
> > >
> > > |Model|FID|Wallclock Time (single sample)|
> > > |-|-|-|
> > > |RQ-Transformer|5.50|5.35s|
> > > |AR-ResGen (num_iter=1)|6.47|1.30s|
> > > |AR-ResGen (num_iter=2)|5.33|1.56s|
> > > |AR-ResGen (num_iter=4)|5.30|2.00s|
> > > |AR-ResGen (num_iter=8)|5.22|3.05s|
> > >
> > > These results clearly illustrate that AR-ResGen significantly reduces the generation time compared to RQ-Transformer, achieving a speedup ranging from approximately 1.75× to 4.1×, depending on the number of depth iterations. Notably, AR-ResGen reaches competitive or better FID scores within just two iterations, corresponding to roughly a 3.4× speedup relative to the original RQ-Transformer’s 5.35s sampling time. This explicitly validates our original claim that our cumulative embedding approach to depth modeling provides substantial practical efficiency improvements in terms of both generation speed and quality.
> > >
> > > **Regarding the discrepancy noted by the reviewer in reported VAR’s FID scores:** Table 2 in the manuscript correctly presents the accurate values, whereas Figure 2 displays values from an earlier arXiv version (version 1). We sincerely apologize for this oversight and will correct Figure 2 in the revised manuscript to ensure consistency and clarity.
> > >
> > > **Regarding the efficiency comparison between VAR and ResGen:** While we acknowledge VAR’s faster single-sample generation compared to ResGen, it's crucial to highlight that ResGen provides a significant advantage in terms of maximum batch size, enabling substantially greater throughput and parallelism during generation. From a modeling perspective, ResGen fundamentally differs from VAR, as elaborated in our response to **Reviewer BBR2**’s comment on **"Weakness 2: Elaborating Differences with VAR."**
> > >
> > > We sincerely appreciate the reviewer’s detailed observations, and we will carefully incorporate these clarifications into the revised manuscript.

---

### Official Review · Reviewer_Bd57 · 2025-03-15

**Overall Recommendation:** 3

**Summary:**

This paper proposes ResGen, an efficient RVQ-based generative modeling for balancing quality and efficiency. It involves a masked token modeling strategy similar to MaskGiT, and a multi-token prediction pipeline inspired by CLaM-TTS, in a discrete diffusion process and variational inference. Experimental results demonstrate that ResGen outperforms autoregressive counterparts in both tasks, without compromising sampling speed.

## update after rebuttal
We thank authors' detailed rebuttal. Most of my concerns have been solved. I would like to improve the rate.

**Claims And Evidence:**

Claim: ResGen outperforms autoregressive counterparts in both tasks, without compromising sampling speed.

Question: In Tab. 1, ResGen shows a slightly higher FID score compared to MAR-L, achieving comparable quality (1.93 of ResGen v.s. 1.78 of MAR-L) with a much faster sampling speed. However, MAR [autoregressive’24] by Li et al. of NeurIPS 2024 demonstrates the inference time of MAR-L about 0.3 sec/image, which is significantly faster than the wallclock time over 100 sec in this paper in Fig. 2. Moreover, MAR with lower steps is also sufficient to achieve a strong generation quality. Could ResGen achieve higher performance compared with a faster MAR with lower steps?

Potential Improvement: It will be helpful to figure out the details of the experiment environment, especially the GPU type and AR steps used in MAR. Further evaluation of MAR with lower steps could also be beneficial for proving the necessity of using ResGen.

**Essential References Not Discussed:**

Most of the references have been discussed.

**Experimental Designs Or Analyses:**

Strengths:
1.	They run on both visual and audio tasks: ImageNet 256×256, and zero-shot TTS tasks.
2.	They compare memory usage (max batch size) and speed to a variety of generative baselines.
3.	They show ablations of top-p, number of steps, and temperature in the appendix.

Potential Weaknesses:
1.	In Tab. 1, RQ-Transformer uses rejection sampling rather than CFG, which might be misleading.
2.	In Tab. 1 and Fig. 2, The speed comparisons and maximum batch size comparisons are primarily self-reported. Without consistent inference setups or more hardware details, we can’t be sure these speed gains generalize.
3.	Comparing with the conventional RQ-transformer which is an autoregression architecture, it is not clear that the preliminary improvement is from the discrete diffusion process or the masking and prediction strategy. A comparison between using autoregressive generation and discrete diffusion generation could help clarify this, similar to MAR [autoregressive’24] by Li et al. of NeurIPS 2024.

**Methods And Evaluation Criteria:**

Methods:
The motivation of this method is to eliminate the problem of sampling complexity associated with sequence length and depth for efficient RVQ-based image generation. The proposed masking and prediction strategy makes sense for the problem. And the discrete diffusion process helps with high-fidelity generation.

Evaluation Criteria:
For image generation,  the paper relies on standard generative modeling metrics like FID, and evaluates the efficiency using metrics such as inference time and batch size. For audio tasks, the paper follows VALL-E (Wang et al., 2023) and CLaM-TTS (Kim et al., 2024).

Potential Improvement:
1.	Better to figure out the details of the experiment environment, especially when evaluating the maximum batch size and inference time.

**Other Comments Or Suggestions:**

Codebook usage details: Detailing the usage of the codebook, especially at each depth, could help in understanding how the masking and prediction strategy and RVQ process work.

**Other Strengths And Weaknesses:**

Strengths:
1.	Take advantage of RVQ and address the computational cost problem, bringing residual codebook design back for fast and high-fidelity image generation.

Additional Weaknesses (detailed):
1.	Complex Implementation: The usage of a mixture of Gaussians for each token plus multi-depth unmasking is quite complicated. The authors do not deeply address computational overhead or memory usage for these mixture parameters, especially as code dimension grows.
2.	Unclear Large-Scale Scaling: Since the authors use depth 16, fewer than the RVQ depth of 32, it remains unclear if that scaling continues to yield benefits or stable training for the performance.

**Questions For Authors:**

1.	Generation Scaling Law: How does ResGen perform when scaling up from 574M for image generation? Could deeper depth gain more from a larger model with more parameters?
2.	Mixture-of-Gaussians Overhead: Since each token position has multiple mixture components, how big is the compute overhead compared to simpler single-Gaussian or discrete classification heads, especially for wide embeddings?

**Relation To Broader Scientific Literature:**

No more specific contributions to a broader scientific literature.

**Theoretical Claims:**

This paper do not have some theoretical claims.

---

> ### Author Rebuttal · Authors · 2025-04-01
>
> We sincerely thank the reviewer for their detailed and constructive feedback. We appreciate the recognition of ResGen's motivation and its potential to address computational costs in RVQ-based generation. We address the specific concerns and questions below:
>
> **Concerning the fair comparison with MAR and further comparison with faster MAR (Claim Question)**, there's a key difference in how speed is measured. MAR reports throughput (sec/image averaged over a large batch, e.g., 256), while we report wall-clock time for single image generation on one A100 GPU. This accounts for the apparent discrepancy (0.3s vs. >100s). To directly address whether ResGen outperforms faster MAR variants, we conducted additional experiments:
>
> 1.  **Reducing MAR's AR Steps (Diffusion=100):** Even MAR-B at 16 AR steps (fastest tested, ~5.6s/image) yielded worse FID than ResGen-rvq16 (FID 4.11 vs. 1.93).
>     |MAR FID (Diffusion=100)|AR=64|AR=32|AR=16|
>     |:-|:-|:-|:-|
>     |MAR-B|2.33|2.44|4.11|
>     |MAR-L|1.81|2.10|4.32|
>
> 2.  **Reducing MAR's Diffusion Steps (AR=256):** MAR-B at 25 diffusion steps (~22.5s/image) performed worse than ResGen-rvq16 (FID 3.38 vs. 1.93).
>     |MAR FID (AR=256)|Diff=100|Diff=50|Diff=25|
>     |:-|:-|:-|:-|
>     |MAR-B|2.31|2.39|3.38|
>     |MAR-L|1.78|1.83|2.22|
>
> These results demonstrate ResGen's consistent advantage in both speed and quality, even against accelerated MAR configurations. We will add this comprehensive comparison to the revision.
>
> **To clarify the experimental environment (Potential Improvement 1, W2.2):** All ResGen evaluations used a single NVIDIA A100 GPU. Training details are in the appendix. For Table 1, we reported the best performance for each model based on a hyperparameter search. Detailed sampling configurations and hyperparameters will be explicitly included in the final manuscript. For Table 2, max batch size was the largest fitting on one A100 using checkpoints' best CFG settings. Inference speed is measured as the wall-clock time required to generate a single sample. We will explicitly state these details in the final manuscript.
>
> **Regarding RQ-Transformer guidance in Table 1 (W2.1):** Thank you for noting potential confusion. For the ablation in Table 1 comparing generation on 8x8x16 tokens, we used CFG for ResGen, RQ-Transformer, and MaskGiT to ensure a fair comparison under identical conditions. As the reviewer noted, the RQ-Transformer paper uses rejection sampling. Rejection sampling is applicable to any generative model, but was not used in Table 1. We will clarify it in the revision.
>
> **Concerning scaling to deeper RVQ depths (Additional Weakness 2):** For images, we used depths 8 and 16 due to excellent reconstruction fidelity (RFID 1.29 and 0.67, Appendix A.4). However, in audio (Table 3), we extensively evaluated deeper depths (32, 72) and observed consistent performance improvements, demonstrating stable training and benefits from increased depth in that domain. This suggests potential benefits for vision too, though not explored here due to already high fidelity at depth 16.
>
> **Regarding clarifying the benefit of discrete diffusion over autoregressive generation:** Please refer to the newly added experiments and analyses presented in our response to **Reviewer aQEq** ("Regarding W1: Isolating the Efficiency Gains of Depth Modeling"), which directly address this point.
>
> **Regarding generation scaling law (Q1):** We investigated scaling ResGen-rvq16 from 574M to 1B parameters on images (400K iterations, batch 256 on 4 GPUs). We observed consistent FID improvements across sampling steps:
>
> |FID (Exp Sampling)|64 steps|48 steps|28 steps|
> |:-|:-|:-|:-|
> |**w/o CFG 574M**|32.44|33.12|33.92|
> |**w/o CFG 1B**|26.58|26.74|28.39|
> |**w/ CFG 574M**|9.53|9.72|9.98|
> |**w/ CFG 1B**|**8.07**|**8.10**|**8.82**|
> This indicates a positive scaling trend, which we will add to the manuscript.
>
> **Regarding the complexity and overhead of the Mixture of Gaussians (MoG) head (Additional Weakness 1, Q2)**, we acknowledge the need for clarity. The MoG head's computational cost is manageable. As detailed in Appendix A.2, parameter prediction involves projecting the hidden output (size `O`) to `K` mixture probabilities, `K` mean vectors (size `H`), and affine parameters. This leads to a projection complexity dominated by `O(O*K*H)`. For our vision model (`O=1152, K=1024, H=64`), this cost is comparable to a standard softmax layer with a ~64K vocabulary (`V = K*H`), which is practical. For higher-dimensional embeddings like in audio (`H=512`), we use low-rank projection for the means (`μ = M * μ_tilde + s`, `h << H`), reducing the dominant computational term to `O(O*K*h + H*h)`, again making the effective cost similar to a ~64K vocabulary model (`h=64`). This technique, previously used in CLaM-TTS, significantly mitigates overhead. We recognize these details were mainly in the Appendix and will clarify the MoG parameterization, its computational cost, and the low-rank optimization in the main paper.

---

### Official Review · Reviewer_q3hF · 2025-03-19

**Overall Recommendation:** 3

**Summary:**

This paper introduces ResGen, an efficient generative modeling method that uses Residual Vector Quantization (RVQ) for high-fidelity data generation. Its key innovation lies in predicting collective token embeddings rather than individual tokens, which decouples inference complexity from quantization depth.

**Claims And Evidence:**

The claims made in the paper—namely, improved efficiency and generation fidelity of RVQ-based generative modeling—are generally well-supported by clear evidence presented through extensive experiments.

**Essential References Not Discussed:**

No significant omissions were noted.

**Experimental Designs Or Analyses:**

The experimental design is sound and well-executed. The authors clearly outline training configurations, baseline comparisons, and evaluation metrics. Ablation studies further strengthen the robustness of their conclusions.

**Methods And Evaluation Criteria:**

The methods and evaluation criteria are well-suited to the tasks at hand.

**Other Comments Or Suggestions:**

None

**Other Strengths And Weaknesses:**

**Strengths:**

- The method introduces a valuable conceptual innovation—predicting collective embeddings to decouple inference complexity from RVQ depth.
- Comprehensive and clear supplementary material strengthens transparency and reproducibility.

**Weaknesses:**

- While extensive, evaluations are limited to specific datasets (ImageNet, standard TTS benchmarks). Performance on more diverse datasets or more challenging, higher-resolution settings could be further explored to validate scalability.
- The paper would benefit from deeper analysis of the results in Table 2, particularly regarding the substantial performance gap between MAR-B and ResGen-rvq16, given that MAR-B has fewer parameters.

**Questions For Authors:**

How sensitive is ResGen's performance to different masking strategies, and could alternative masking approaches significantly affect the results?

**Relation To Broader Scientific Literature:**

The paper fits well within the broader context of generative modeling literature. It makes a significant advance in RVQ-based generative approaches by solving efficiency bottlenecks that arise from depth-dependent inference complexity.

**Theoretical Claims:**

The paper formulates a probabilistic framework. While the paper presents equations clearly, no explicit mathematical proofs requiring detailed checking are presented.

---

> ### Author Rebuttal · Authors · 2025-04-01
>
> We sincerely thank the reviewer for their thorough assessment, positive feedback on our claims, methodology, and supplementary material, and for recognizing the conceptual innovation of ResGen. We appreciate the constructive suggestions for further improvement.
>
> Regarding the points raised:
>
> **Limited Dataset Diversity/Scalability (W1):** We appreciate the suggestion to evaluate ResGen on more diverse datasets and higher-resolution settings. Our current study focused on established benchmarks like ImageNet and standard TTS datasets to ensure the adaptability of this method across different domains. We agree that broader validation is important and plan to explore ResGen's scalability and generalization on more diverse tasks, including higher-resolution image generation, as a key direction for future work. But to provide additional context on model scalability, we conducted scaling experiments with ResGen-rvq16, increasing model parameters from 574M to 1B on ImageNet. Although these scaling experiments are ongoing (currently at 400K iterations, batch size 256 on 4 GPUs), intermediate results reveal FID improvements, as shown below:
> | FID (Exp Sampling) | 64 steps | 48 steps | 28 steps |
> | :----------------- | :------- | :------- | :------- |
> | **w/o CFG 574M**   | 32.44    | 33.12    | 33.92    |
> | **w/o CFG 1B**     | 26.58    | 26.74    | 28.39    |
> | **w/ CFG 574M**    | 9.53     | 9.72     | 9.98     |
> | **w/ CFG 1B**      | **8.07** | **8.10** | **8.82** |
> We will include the fully converged results in the manuscript.
>
> **Analysis of MAR-B vs. ResGen-rvq16 Performance (W2):** Thank you for prompting a deeper analysis of the results in Table 2. While MAR-B has fewer parameters and achieves slightly better FID *without* classifier-free guidance (CFG), ResGen-rvq16 excels *with* CFG. This difference stems from their distinct generative mechanisms. MAR-B utilizes multi-token prediction followed by continuous diffusion steps, allowing iterative refinement and revision of previously generated tokens. In contrast, ResGen performs disjoint token unmasking at each step without revisiting prior decisions. This fundamental difference in modeling leads to varying performance characteristics, particularly highlighted by the impact of CFG. Furthermore, although MAR-B can perform well without CFG, achieving its best results often requires significantly more sampling steps (e.g., 100+ diffusion steps) compared to ResGen, impacting inference speed. Thus, there is a trade-off between MAR-B's potential parameter efficiency (in certain configurations) and ResGen's inference efficiency, especially when aiming for high-quality results with guidance. We will incorporate this more detailed comparative analysis into the revised manuscript.
>
> **Sensitivity to Masking Strategies (Q1):** This is an excellent question regarding the robustness of our approach. We investigated ResGen's sensitivity by training ResGen-rvq16 models (400K iterations) using three distinct masking schedules during training: circle, exponential, and cosine. We then evaluated these models under two sampling conditions: using the same or different  schedule during sampling than training for all models.
>
> Our results, summarized below (FID scores, lower is better), show interesting interactions:
>
> | CFG     | Training | Sampling | 64 steps | 48 steps | 28 steps |
> | :------ | :------- | :------- | :------- | :------- | :------- |
> | **w/o** | cosine   | cosine   | 28.13    | 28.01    | 29.18    |
> | **w/o** | cosine   | exp      | 32.30    | 32.70    | 32.81    |
> | **w/o** | circle   | circle   | 26.04    | 26.41    | 26.73    |
> | **w/o** | circle   | exp      | 32.44    | 33.12    | 33.92    |
> | **w/o** | exp      | exp      | 41.12    | 41.67    | 41.87    |
> | **w/**  | cosine   | cosine   | 15.46    | 15.69    | 17.27    |
> | **w/**  | cosine   | exp      | 9.66     | 9.78     | 10.08    |
> | **w/**  | circle   | circle   | 10.09    | 10.35    | 10.44    |
> | **w/**  | circle   | **exp**  | **9.53** | **9.72** | **9.98** |
> | **w/**  | exp      | exp      | 12.63    | 12.65    | 12.83    |
>
> Notably, the best performance (FID 9.53 at 64 steps with CFG) was achieved when training with the *circle* masking strategy but sampling using the *exponential* strategy. This suggests that the different training and sampling schedules impact the final quality. The exponential schedule unmasks fewer tokens in the crucial early stages and more tokens later. This coarse-to-fine unmasking during inference appears beneficial for ResGen, likely allowing the model to establish a more stable initial prediction before revealing finer details. The strong performance under exponential sampling, even when trained differently, indicates its effectiveness for inference with ResGen. We will include a detailed discussion of these findings in the revised manuscript.

---

### Official Review · Reviewer_BBR2 · 2025-03-19

**Overall Recommendation:** 4

**Summary:**

This paper introduces ResGen, an efficient generative model leveraging residual vector quantization (RVQ). While RVQ typically enhances image fidelity by increasing quantization depth, it also demands more inference steps during sampling. Instead of sequentially predicting tokens at each depth, ResGen proposes a novel approach: predicting the sum of masked tokens at each layer, known as cumulative tokens. These cumulative tokens are then re-quantized using RVQ. By effectively integrating MaskGit with RVQ, ResGen achieves impressive performance improvements.

**Claims And Evidence:**

Yes

**Essential References Not Discussed:**

HART: Efficient Visual Generation with Hybrid Autoregressive Transformer. [ICLR'25]

**Experimental Designs Or Analyses:**

Yes, no issues.

**Methods And Evaluation Criteria:**

Make sense

**Other Comments Or Suggestions:**

N/A

**Other Strengths And Weaknesses:**

Strengths:
1. ResGen effectively combines MaskGit and RVQ for efficient generative modeling.
2. ResGen achieve significant performance improvements over MaskGit and efficiency improvement over RQ-Transformer.
3. The author provides the theoretical justification for RVQ.

Weaknesses:
1. Will the inference step be further decreased for faster inference?
2. The discussion on the differences between ResGen and VAR could be elaborated. Since VAR also considers the hierarchical depth in RVQ and formulates it as a scale, a more detailed comparison would better highlight ResGen’s advantages and solidify its contributions.

**Questions For Authors:**

N/A

**Relation To Broader Scientific Literature:**

All appendix

**Theoretical Claims:**

I check the Section 3.2.
In lines 263-265, Shouldn't p(x^(0)|z, x^(t)) corresponds to RVQ dequantization instead of quantization? It seems there might be a mix-up here. Could you clarify?

---

> ### Author Rebuttal · Authors · 2025-04-01
>
> We sincerely thank the reviewer for the positive evaluation and constructive feedback.
>
> We address the specific points raised below:
>
> **Essential References Not Discussed (HART):**
> Thank you for bringing HART to our attention. We agree it is a relevant recent work and will incorporate a discussion into our Related Work section. HART proposes a hybrid approach, distinct from purely discrete tokenizers like ours (ResGen, VAR) or continuous ones (MAR, GIVT). It decomposes latents into discrete tokens (modeled autoregressively) and continuous residuals (modeled via diffusion). This hybrid strategy aims to reduce sampling steps compared to continuous methods like MAR.
>
> **Theoretical Claims (Sec 3.2, Lines 263-265):**
> Thank you for the careful reading and request for clarification. There might be slight confusion in terminology depending on perspective (encoding vs. decoding of VQ). Let us clarify our notation:
> *   `z` represents the continuous cumulative embeddings corresponding to the target clean tokens `x^(0)`. Hence the term `q(z | x^(0), x^(t))` represents a form of dequantization or embedding lookup.
> *   `p(x^(0)|z, x^(t))` represents the inference of the true clean tokens `x^(0)` given this embeddings `z` and the current masked tokens `x^(t)`. This involves finding the discrete tokens `x^(0)` whose corresponding embeddings are `z`. This step effectively performs the **quantization** of the continuous embedding `z` back into the discrete RVQ code space.
>
> So, `p(x^(0)|z, x^(t))` indeed relates to determining the discrete tokens `x^(0)` *from* the continuous embedding `z`, which involves the RVQ quantization mechanism applied to `z`.
>
> **Weakness 1: Potential for Further Inference Speed-up:**
> Thank you for this suggestion. In the vision domain, reducing sampling steps to 18 (Appendix B.2) improves inference speed but slightly decreases generation quality (FID 3.94). However, experiments in the audio domain show that fewer steps (8 and 16 steps) still achieve comparable performance (see Table below).
> |Continuation|WER|CER|SIM-o|SIM-r|
> |-|-|-|-|-|
> | melvae-resgen-25step|1.94|0.53|0.5421|0.5701|
> | melvae-resgen-16step|1.92|0.53|0.5419|0.5705|
> | melvae-resgen-8step|1.92|0.54|0.5429|0.5710|
> | rvqvae-resgen-25step|1.86|0.50|0.5853|0.5886|
> | rvqvae-resgen-16step|1.87|0.52|0.5820|0.5864|
> | rvqvae-resgen-8step|1.86|0.51|0.5847|0.5886|
>
> |Cross|WER|CER|SIM-o|SIM-r|
> |-|-|-|-|-|
> |melvae-resgen-25step|1.75|0.48|0.5597|0.6061|
> |melvae-resgen-16step|1.92|0.53|0.5419|0.5705|
> |melvae-resgen-8step|1.93|0.54|0.5433|0.5708|
> |rvqvae-resgen-25step|1.70|0.46|0.6037|0.6307|
> |rvqvae-resgen-16step|1.75|0.46|0.6037|0.6302|
> |rvqvae-resgen-8step|1.95|0.50|0.5898|0.6108|
>
>
> **Weakness 2: Elaborating Differences with VAR:**
> Thank you for this suggestion; a clearer comparison with VAR will indeed strengthen the paper. While both ResGen and VAR utilize the hierarchical nature of RVQ, they differ:
> *   **Structure & Resolution:** VAR assigns different spatial resolutions to different RVQ depths (e.g., 1x1, 2x2, ... up to original resolution), requiring a predefined hierarchy. RVQ uses all quantization depths to refine the representation at a *single* resolution which is identical to the length of output sequence of the VAE encoder.
> *   **Generative Process:** VAR generates tokens autoregressively across depths, although sampling within a depth can be parallel given the previous depth. ResGen predicts *cumulative embeddings* representing multiple depths simultaneously within a masked generation framework, allowing parallel prediction across the sequence length dimension.
> *   **Task Adaptability:** VAR's predefined resolution hierarchy might be less straightforward to adapt to tasks with arbitrary output lengths, such as text-to-speech, where ResGen is easily applied.
> *   **Resolution Flexibility:** RVQ allows us to achieve lower final spatial resolutions (e.g., 8x8) by increasing quantization depth (e.g., 16), offering flexibility in balancing sequence length and depth. Achieving similarly low spatial resolution might be less direct in VAR's depth-resolution coupled structure.
>
> We will incorporate a more detailed discussion contrasting these aspects in the related work.

---

### Official Review · Reviewer_nBTE · 2025-03-23

**Overall Recommendation:** 3

**Summary:**

This paper introduces ResGen, a method that directly predicts vector embeddings for groups of tokens rather than individual tokens. This design reduces the number of inference steps, thereby improving latency. Token masking is employed during training, while multi-token prediction is utilized during inference. Experimental results on image generation and audio synthesis demonstrate the effectiveness of the proposed approach.
## update after rebuttal
Thanks for the authors rebuttal. Most of my concerns have been addressed and I decide to increase my rating.

**Claims And Evidence:**

See Strength and Weakness.

**Essential References Not Discussed:**

N/A.

**Experimental Designs Or Analyses:**

Yes.

**Methods And Evaluation Criteria:**

See Strength and Weakness.

**Other Comments Or Suggestions:**

Please kindly review the weaknesses I have outlined and provide clarifications for each point individually.

**Other Strengths And Weaknesses:**

Strength:

- By eliminating the auto-regressive prediction of depth tokens, the proposed method improve the inference latency.

Weakness and Questions:

- **Method**: My main concern is that the proposed method predicts continuous embeddings rather than discrete tokens, subsequently quantizing these embeddings into discrete tokens. Given this approach, it's unclear why the method doesn't simply predict continuous embeddings directly, similar to diffusion methods. The quantization step inevitably introduces information loss, raising questions about the purpose and effectiveness of using discrete tokens.

- **Experiment**: In Section 5.2.1, the paper presents a reimplementation of MaskGiT using RVQ quantization with a depth of 16. However, the implementation details are unclear. If the reimplementation yields significantly worse results compared to the original MaskGiT—as Table 1 suggests—then its utility as a baseline becomes questionable.

- **Expression**: Throughout the paper, the definition of 'discrete diffusion model' is unclear. Specifically, methods such as MaskGIT, VAR, and RQ-VAE, which are listed in the related works, do not align with what I would consider discrete diffusion models.

**Questions For Authors:**

See Weakness.

**Relation To Broader Scientific Literature:**

N/A.

**Theoretical Claims:**

Yes.

---

> ### Author Rebuttal · Authors · 2025-04-01
>
> We thank the reviewer for their insightful comments, which help improve our work's clarity. We address each point below:
>
> \
> **Q1: Rationale for Predicting Quantized Discrete Tokens vs. Continuous Embeddings**
>
> Our approach of predicting discrete RVQ tokens via cumulative embeddings offers significant inference efficiency advantages over directly predicting high-dimensional continuous embeddings.
>
> While direct continuous prediction (e.g., DiT) is valid, it often requires many inference steps. Our method targets discrete RVQ tokens (8x8x16) for efficiency, even though it uses continuous cumulative embeddings internally.
>
> Generating masked tokens generally demands fewer inference steps than refining high-dimensional continuous vectors, as shown by MaskGiT achieving strong results faster than many continuous methods. Quantization, despite information loss, simplifies the target space, reducing the burden of precisely predicting large continuous vectors (e.g., 8x8x64 in our case).
>
> Predicting cumulative embeddings enables a few step iterative prediction of all 16 token depths. This contrasts with single-step embedding prediction (e.g., GIVT) or continuous diffusion models needing many refinement steps. Quantization is crucial for this multi-depth strategy's feasibility.
>
> In essence, we leverage internal continuous representations for modeling flexibility but target discrete tokens for efficient inference, particularly suited for our multi-depth prediction approach.
>
> \
> **Q2: MaskGiT Reimplementation Baseline in Ablation Study (Section 5.2.1)**
>
> The MaskGiT variant in Table 1 uses 8x8x16 RVQ tokens specifically for a controlled comparison within our ablation study on multi-depth token generation, not to replicate the original MaskGiT's performance on its native 16x16x1 format. The experiments aimed to compare strategies for generating multi-depth RVQ tokens (8x8x16): (i) fully autoregressive (RQ-Transformer), (ii) masked sequence + autoregressive depth (our MaskGiT variant), and (iii) fully masked (ResGen). Fair comparison required all models to operate on the same 8x8x16 RVQ format. The relatively lower performance of the MaskGiT variant (predicting depth-by-depth) compared to ResGen highlights the challenge of extending single-depth masked models to multi-depth RVQ tokens and supports our approach of predicting all depths collectively.
>
> To further strengthen this analysis and provide a more precise ablation, we conducted additional experiments:
> 1.  As detailed in our response to Reviewer aQEq (W1), we implemented an "AR-ResGen" model which combines an autoregressive sequence model (like RQ-Transformer's) with ResGen's efficient iterative depth prediction using cumulative embeddings. This isolates the benefit of our depth modeling strategy.
> 2.  We also implemented a variant of our method which uses the same masked generation framework introduced in our method section but predicts discrete tokens directly and in parallel across all depths at each step, *instead* of predicting continuous cumulative embeddings. This represents a more direct application of masked diffusion to RVQ depths. This variant achieved FID scores of **12.79 (w/o CFG)** and **2.91 (w/ CFG)**.
>
> These additional experiments demonstrate that while variations of our method can perform well, our proposed final method using cumulative vector embedding prediction achieves the best overall performance within our ablation study, validating its effectiveness.
>
> We will revise Section 5.2.1 to clarify the baseline's purpose (controlled comparison on 8x8x16 RVQ tokens), implementation (sequential depth prediction), its role within the ablation, and add the results of our new ablation experiments as requested.
>
>
> \
> **Q3: Clarity of the Term 'Discrete Diffusion Model'**
>
> We acknowledge the ambiguity regarding 'discrete diffusion model' and thank you for highlighting it.
>
> Our discussion of 'discrete diffusion models' in the related works primarily referred to models like VQ-Diffusion, GIVT, and conceptually MaskGIT, which learn to reverse a corruption process (like masking) on discrete data (tokens). This was mainly confined to the first paragraph of the related works.
>
> Other mentioned models (e.g., VAR, RQ-Transformer) were included for broader context in token-based modeling but were not necessarily classified by us as discrete diffusion models.
>
> We also aimed to highlight 'masked diffusion' (corruption via masking only) as an effective subset of the discrete diffusion model, following D3PM and VQ-Diffusion.
>
> We will revise the paper to provide a precise discrete diffusion model definition as used, clearly delineate this category from related models, explicitly refer masked diffusion, and ensure consistent terminology throughout.
>
> \
> We hope these clarifications and the planned revisions adequately address the reviewer's concerns. We appreciate the constructive feedback.

---

### Decision · Program_Chairs · 2025-05-01

**Decision:**

Accept (poster)

**Comment:**

This paper introduces an efficient RVQ-based generative model for fast sampling by predicting vector embeddings for groups of tokens. It received mixed reviews initially, where reviewers have concerns regarding motivations, differences with prior work, and missing partial comparisons with competing methods.

After the rebuttal, the authors addressed most of the concerns, and three reviewers increased their final ratings. By taking the overall contributions and additional results in the rebuttal into consideration, AC believes most significant concerns have been alleviated and this paper is above the acceptance threshold.